# SUFFICIENT SUBGRAPH EMBEDDING MEMORY FOR CONTINUAL GRAPH REPRESENTATION LEARNING

## ABSTRACT

Memory replay, which constructs a buffer to store representative samples and retrain the model over the buffer to maintain its performance over existing tasks, has shown great success for continual learning with Euclidean data. Directly applying it to graph data, however, can lead to the memory explosion problem due to the necessity to consider explicit topological connections of representative nodes. To this end, we present Parameter Decoupled Graph Neural Networks (PDGNNs) with Sufficient Subgraph Embedding Memory (SSEM) to fully utilize the explicit topological information for memory replay and reduce the memory space complexity from $\mathcal{O}(nd^L)$ to $\mathcal{O}(n)$, where $n$ is the memory buffer size, $d$ is the average node degree, and $L$ is the range of neighborhood aggregation. Specifically, PDGNNs decouple trainable parameters from the computation subgraphs via *Sufficient Subgraph Embeddings* (SSEs), which compress subgraphs into vectors (*i.e.*, SSEs) to reduce the memory consumption. Besides, we discover a *pseudo-training effect* in memory based continual graph learning, which does not exist in continual learning on Euclidean data without topological connection (*e.g.*, individual images). Based on the discovery, we develop a novel *coverage maximization sampling* strategy to enhance the performance when the memory budget is tight. Thorough empirical studies demonstrate that PDGNNs with SSEM outperform state-of-the-art techniques for both class-incremental and task-incremental settings.

## 1 INTRODUCTION

Continual graph representation learning (Liu et al., 2021; Zhou & Cao, 2021; Zhang et al., 2021), which aims to accommodate new types of emerging nodes in a graph and their associated edges without interfering with the model performance over existing nodes, is an emerging area that attracts increasingly more attention recently. It exhibits enormous value in various practical applications, especially in the case where graphs are relatively large and retraining a new model over the entire graph is computationally infeasible. For instance, in a social network, a community detection model has to keep adapting its parameters based on nodes from newly emerged communities; in a citation network, a document classifier needs to continuously update its parameters to distinguish the documents of newly emerged research fields.

Memory replay (Rebuffi et al., 2017; Lopez-Paz & Ranzato, 2017; Aljundi et al., 2019; Shin et al., 2017), which stores representative samples in a buffer for retraining the model to maintain its performance over existing tasks, exhibits great success in preventing catastrophic forgetting for various continual learning tasks, *e.g.*, computer vision and reinforcement learning (Kirkpatrick et al., 2017; Li & Hoiem, 2017; Aljundi et al., 2018; Rusu et al., 2016). Directly applying memory replay to graph data with message passing based graph neural networks (GNNs) (Gilmer et al., 2017; Kipf & Welling, 2016; Veličković et al., 2017), however, could give rise to the memory explosion problem. Specifically, due to the message passing over the topological connections in graphs, retraining an $L$-layer GNN (Figure 1 a) with $n$ buffered nodes would require storing $\mathcal{O}(nd^L)$ nodes (Chiang et al., 2019; Chen et al., 2017) (the number of edges is not counted yet) in the buffer, where $d$ is the average node degree. Take the Reddit dataset (Hamilton et al., 2017) for an example, its average node degree is 492, the buffer size will easily be intractable even with a 2 layer GNN. To overcome this issue, Experience Replay based GNN (ER-GNN) (Zhou & Cao, 2021) stores representative nodes in the buffer but completely ignores the topological information (Figure 1 b). Feature graph network (FGN) (Wang et al., 2020a) implicitly encodes node proximity with the inner products between the features of the target node and its neighbors. However, the explicit topological connections are completely ignored and message passing is no longer feasible on the graph.

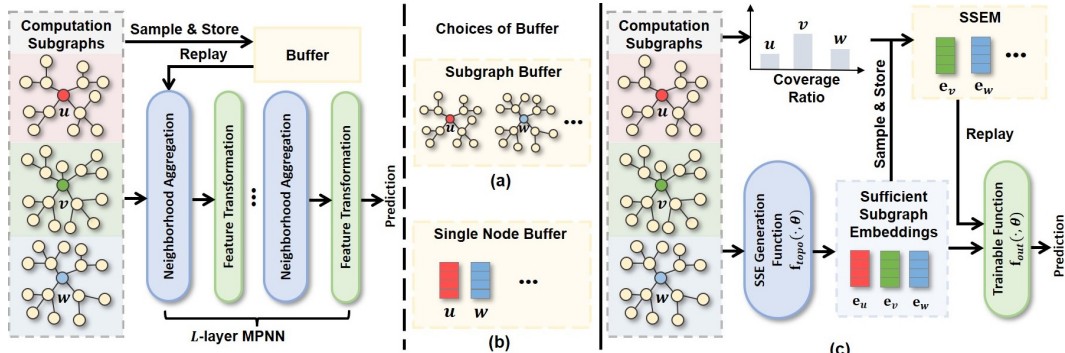

Figure 1: (a) Directly storing computation subgraphs for replay in a multi-layer MPNN. (b) The strategy to store single nodes proposed in ER-GNN (Zhou & Cao, 2021). (c) Our PDGNNs with SSEM. The incoming computation subgraphs are first embedded as SSEs and then fed into the trainable function. The SSEs are sampled and stored with the probability computed based on their coverage ratio, *i.e.,* the ratio of nodes covered by their computation subgraphs (Section 3.6).

To this end, we propose Parameter Decoupled Graph Neural Networks (PDGNNs) with Sufficient Subgraph Embedding Memory (SSEM) for continual graph learning. Since the key challenge lies in the unbounded sizes of the computation subgraphs, we introduce the concept of *Sufficient Subgraph Embedding* (SSE) with a fixed size but contains all necessary information of a computation subgraph for model optimization. Such SSEs can be surrogates of computation subgraphs in memory replay. Next, we found that it is infeasible to derive SSEs from MPNNs since their trainable parameters and individual nodes/edges are entangled. To this end, we formulate the PDGNNs framework to decouple them and enable memory replay only based on buffered SSEs (without the computation subgraphs). Since the size of an SSE is fixed, the memory space complexity of a buffer with size $n$ can be dramatically reduced from $\mathcal{O}(nd^L)$ to $\mathcal{O}(n)$. Moreover, different from traditional continual learning on data without topology (*e.g.*, images), we discover that replaying an SSE incurs a *pseudo-training effect* on the neighbor nodes, which strengthens the prediction of the other nodes in the same computation subgraph. This effect is unique in continual graph learning and takes place due to the neighborhood aggregation in GNNs. We further analyze that in homophilous graphs (prevalent in real-world data), the pseudo-training effect makes the SSEs corresponding to larger computation subgraphs (quantitatively measured by coverage ratio) more beneficial to continual learning. Inspired by this, we develop a novel *coverage maximization sampling*, which enlarges the coverage ratio of the selected SSEs and empirically enhances the performance without consuming additional memory. In experiments, we adopt both the class-incremental (class-IL) continual learning scenario (Rebuffi et al., 2017) (rarely studied for node classification under the continual learning setting) and the task-incremental (task-IL) scenario (Liu et al., 2021; Zhou & Cao, 2021). Thorough empirical studies demonstrate that PDGNNs with SSEM outperform state-of-the-art continual graph representation learning techniques for both class-IL and task-IL settings. Our contributions are summarized below:

- We formulate the framework of PDGNNs-SSEM, which successfully enable memory replay with topological information for continual graph representation learning, and reduce the memory space complexity from $\mathcal{O}(nd^L)$ to $\mathcal{O}(n)$.
- PDGNNs-SSEM obtain superior performance especially in the challenging class-IL scenario.
- We theoretically reveal a unique phenomenon in continual graph learning (*i.e.* the pseudo-training effect) when applying memory replay, and accordingly develop the *coverage maximization sampling* strategy to leverage this effect for improving the performance.

## 2 RELATED WORKS

Our proposed PDGNNs-SSEM is closely related to continual learning, continual graph learning, and decoupled graph neural networks.

### 2.1 CONTINUAL LEARNING & CONTINUAL GRAPH LEARNING

To alleviate the catastrophic forgetting problem encountered by machine learning models, *i.e.,* drastic performance decrease on previous tasks after learning new tasks, existing approaches can be categorized into three types. Regularization based methods apply different constraints to prevent drastic modification of model parameters that are important for previous tasks (Farajtabar et al., 2020; Kirkpatrick et al., 2017; Li & Hoiem, 2017; Aljundi et al., 2018; Hayes & Kanan, 2020). Parametric isolation methods adaptively allocate new parameters for the new tasks to protect the parameters for

the previous tasks (Wortsman et al., 2020; Wu et al., 2019b; Yoon et al., 2020; 2017; Rusu et al., 2016). Memory replay based methods alleviate forgetting by storing and replaying representative data examples from previous tasks when learning new tasks (Caccia et al., 2020; Chrysakis & Moens, 2020; Rebuffi et al., 2017; Lopez-Paz & Ranzato, 2017; Aljundi et al., 2019; Shin et al., 2017). Recently, continual learning on graphs attracts increasingly more attention due to its practical importance (Zhou & Cao, 2021; Zhang et al., 2021; Liu et al., 2021; Wang et al., 2020b; Xu et al., 2020; Daruna et al., 2021). Existing works include regularization methods like topology-aware weight preserving (TWP) (Liu et al., 2021) to preserve crucial parameters and topologies, parametric isolation approaches like HPNs (Zhang et al., 2021) that adaptively select different parameters for different tasks, and memory replay methods like ER-GNN (Zhou & Cao, 2021) that stores representative nodes. Our work is also based on memory replay and its key advantage lies in being capable of preserving complete topological information with reduced space complexity, which shows significant superiority in class-IL setting (Section 4.4). Note that we study the class-IL for node classification, which is essentially different from the class-IL for graph-level prediction (Carta et al., 2021). Memory replay for graph-level tasks stores individual graphs and will not trigger the memory explosion problem (same as traditional continual learning on Euclidean data). In this work, we focus on the class-IL for node classification and aim to resolve the memory explosion problem. Finally, it is worth highlighting the difference between continual graph learning and some relevant but different research areas. First, dynamic graph learning (Galke et al., 2020; Wang et al., 2020c; Han et al., 2020; Yu et al., 2018; Nguyen et al., 2018; Zhou et al., 2018; Ma et al., 2020; Feng et al., 2020) focuses on the temporal node dynamics with all previous data being accessible. In contrast, continual graph learning aims to alleviate forgetting, therefore the previous data is inaccessible. Second, few-shot graph learning (Zhou et al., 2019; Guo et al., 2021; Yao et al., 2020; Tan et al., 2022) targets fast adaptation to new tasks. In training, few-shot learning models can access all previous tasks simultaneously (unavailable in continual learning). For evaluation, few-shot learning models need to be fine-tuned on the new test classes, while the continual learning models are evaluated over existing tasks without fine-tuning.

## 2.2 DECOUPLED GRAPH NEURAL NETWORKS & RESERVOIR COMPUTING

Unlike the early works with interleaved neighborhood aggregation and node feature transformation (Kipf & Welling, 2016; Gilmer et al., 2017; Veličković et al., 2017; Xu et al., 2018; Chen et al., 2018; Hamilton et al., 2017), recent works reveal that decoupling these two operations can reduce complexity and increase scalability, while maintaining equivalent or even achieving superior performance for GNNs (Zeng et al., 2021; Chen et al., 2020; 2019; Nt & Maehara, 2019; Frasca et al., 2020). For instance, Simple Graph Convolution (SGC) (Wu et al., 2019a) removes the non-linear activations from GCN and only keeps one neighborhood aggregation and one node transformation layer. Approximate Personalized Propagation of Neural Predictions (APPNP) (Klicpera et al., 2018) first performs node transformation and then conducts multiple neighborhood aggregations in one layer. Following these works, Dong et al. (2021) prove that the decoupling strategy to predict then propagate is equivalent to training on the unlabelled nodes with pseudo labels aggregated from the labeled neighbors. To further explore decoupled GNNs, Chen et al. (2020) formulate the Graph-Augmented Multi-Layer Perceptrons (GA-MLPs), and theoretically analyzed their expressive power. Instead of decoupling the structures, Zeng et al. (2021) propose SHADOW-GNN to decouple the depth and scope of GNNs by fixing the depth of the computation subgraph. Among these works, some can be viewed as instantiations of PDGNNs (Wu et al., 2019a; Zhu & Koniusz, 2020; Gallicchio & Micheli, 2020), while the others may not focus on decoupling the trainable parameters and the space complexity is still $\mathcal{O}(nd^L)$ when applying memory replay, *e.g.*, APPNP (Klicpera et al., 2018), Propagation then Training Adaptively (PTA) (Dong et al., 2021), *etc.*. Besides works on decoupling GNNs, PDGNNs are also related to reservoir computing based GNNs (Gallicchio & Micheli, 2020; 2010), which embed the graphs via a fixed, non-linear system followed by a trainable linear readout module. The reservoir computing modules can be adopted in PDGNNs as the SSE generation function (Equation 4), and the corresponding experimental results are in Appendix C.5.

# 3 PARAMETER DECOUPLED GNNS WITH SUFFICIENT SUBGRAPH EMBEDDING MEMORY

In this section, we first introduce the notations and then explain the technical challenge of applying memory replay techniques to GNNs. Targeting the challenge, we introduce PDGNNs with *Sufficient Subgraph Embedding Memory* (SSEM). Finally, inspired by theoretical findings of the *pseduo-training effect*, we develop the coverage maximization sampling. It can empirically improve the continual learning performance, especially when the memory budget is tight. All detailed proofs are provided in the Appendix B.

### 3.1 PRELIMINARIES

In this paper, continual graph learning is formulated as learning node representations on a sequence of subgraphs (tasks): $\mathcal{S} = \{\mathcal{G}_1, \mathcal{G}_2, ..., \mathcal{G}_T\}$. Each subgraph $\mathcal{G}_\tau$ contains several new emerging categories of nodes in the overall graph and is associated with a node set $\mathbb{V}_\tau$ and an edge set $\mathbb{E}_\tau$, which is represented as the adjacency matrix $\mathbf{A}_\tau \in \mathbb{R}^{|\mathbb{V}_\tau| \times |\mathbb{V}_\tau|}$. Each entry of $\mathbf{A}_\tau$ denotes an edge between a pair of nodes. The degree of a node $d$ refers to the number of edges connected to it. In practice, $\mathbf{A}_\tau$ is often normalized as $\hat{\mathbf{A}}_\tau = \mathbf{D}_\tau^{-\frac{1}{2}} \mathbf{A}_\tau \mathbf{D}_\tau^{-\frac{1}{2}}$, where $\mathbf{D}_\tau \in \mathbb{R}^{|\mathbb{V}_\tau| \times |\mathbb{V}_\tau|}$ is the degree matrix. Each node $v \in \mathbb{V}_\tau$ has a feature vector $\mathbf{x}_v \in \mathbb{R}^b$. In classification tasks, each node $v$ has a label $\mathbf{y}_v \in \{0, 1\}^C$, where $C$ is the total number of classes. When generating the representation for a target node $v$, GNNs typically take a subgraph within $\mathcal{G}_\tau$ as the input, which is denoted as the computation subgraph $\mathcal{G}_{\tau,v}^{sub}$. For simplicity, $\mathcal{G}_v^{sub}$ may be used in the following, without the graph index. We define the $L$-hop neighbors of a node $v$ as $\mathcal{N}^L(v)$ which contains all nodes within a distance of $L$ from $v$.

### 3.2 MEMORY REPLAY MEETS GNNS

In traditional continual learning, a model $f(\cdot; \boldsymbol{\theta})$ parameterized by $\boldsymbol{\theta}$ is trained on a sequence of $T$ tasks. Each task $\tau$ ($\tau \in \{1, ..., T\}$) corresponds to a dataset $\mathbb{D}_\tau = \{(\mathbf{x}_i, \mathbf{y}_i)_{i=1}^{n_\tau}\}$. To avoid forgetting, memory replay based methods store representative data from the old tasks in a buffer $\mathcal{B}$, which are replayed when learning new tasks. A common approach to utilize $\mathcal{B}$ is through an auxiliary loss:

$$\mathcal{L} = \underbrace{\sum_{\mathbf{x}_i \in \mathbb{D}_\tau} l(f(\mathbf{x}_i; \boldsymbol{\theta}), \mathbf{y}_i)}_{\mathcal{L}_\tau: \text{ loss of the current task}} + \lambda \underbrace{\sum_{\mathbf{x}_j \in \mathcal{B}} l(f(\mathbf{x}_j; \boldsymbol{\theta}), \mathbf{y}_j)}_{\mathcal{L}_{aux}: \text{ auxiliary loss}}, \tag{1}$$

where $l(\cdot, \cdot)$ denotes the loss function, and $\lambda \geq 0$ balances the contribution of the old data. The buffer $\mathcal{B}$ may also be used in other ways to prevent forgetting instead of directly minimizing $\mathcal{L}_{aux}$ Lopez-Paz & Ranzato (2017); Rebuffi et al. (2017). In these applications, the space complexity of a buffer containing $n$ examples is $\mathcal{O}(n)$.

However, to capture the topological information, GNNs obtain the representation of a node $v$ based on a computation subgraph surrounding $v$. We exemplify it with the popular MPNN framework (Gilmer et al., 2017), which updates the hidden node representations at the $l + 1$-th layer as:

$$\mathbf{m}_v^{l+1} = \sum_{w \in \mathcal{N}^1(v)} M_l(\mathbf{h}_v^l, \mathbf{h}_w^l, \mathbf{x}_{v,w}^e; \boldsymbol{\theta}_l^M), \qquad \mathbf{h}_v^{l+1} = U_l(\mathbf{h}_v^l, \mathbf{m}_v^{l+1}; \boldsymbol{\theta}_l^U), \tag{2}$$

where $\mathbf{h}_v^l$, $\mathbf{h}_w^l$ are hidden representations of nodes at layer $l$, $\mathbf{x}_{v,w}^e$ is the edge feature, $M_l(\cdot, \cdot, \cdot; \boldsymbol{\theta}_l^M)$ is the message function to integrate neighborhood information, and $U_l(\cdot, \cdot; \boldsymbol{\theta}_l^U)$ updates $\mathbf{m}_v^{l+1}$ into $\mathbf{h}_v^l$. When $l = 0$, $\mathbf{h}_v^0$ denotes the input node features. In a $L$-layer MPNN, the representation of a node $v$ can be simplified as,

$$\mathbf{h}_v^L = \text{MPNN}(\mathbf{x}_v, \mathcal{G}_v^{sub}; \boldsymbol{\Theta}), \tag{3}$$

where $\mathcal{G}_v^{sub}$ is the computation subgraph containing the $L$-hop neighbors (*i.e.*, $\mathcal{N}^L(v)$), $\text{MPNN}(\cdot, \cdot; \boldsymbol{\Theta})$ is the composition of all $M_l(\cdot, \cdot, \cdot; \boldsymbol{\theta}_l^M)$ and $U_l(\cdot, \cdot; \boldsymbol{\theta}_l^U)$ at different layers. Since $\mathcal{N}^L(v)$ typically contains $\mathcal{O}(d^L)$ nodes, replaying $n$ sampled nodes would require storing $\mathcal{O}(nd^L)$ nodes (the edges of $\mathcal{G}_v^{sub}$ are not counted yet), where $d$ is the average node degree. Take the Reddit dataset (Hamilton et al., 2017) as a concrete example, its average degree is 492, even with a 2 layer MPNN, the buffer size will be easily intractable. Therefore, directly storing the computation subgraphs for memory replay is infeasible for GNNs. Besides, the unsupervised learning models Adhikari et al. (2018); Narayanan et al. (2016) also suffer from this problem. Because the trainable parameters in the unsupervised learning part will also be updated after learning each task, the original computation subgraphs are required for retraining the model.

### 3.3 PARAMETER DECOUPLED GNNS WITH SSEM

As we discussed earlier, the key difficulty of applying memory replay to graph data is to store the computation subgraphs with potentially unbounded sizes. Therefore, we would naturally expect to preserve the necessary information (*e.g.*, the topological information) of a computation subgraph with a vector of fixed length such that the memory consumption can be manageable. Formally, the desired subgraph representation can be defined as *Sufficient Subgraph Embedding* (SSE).

**Definition 1** (Sufficient subgraph embedding). *Given a model parameterized with $\boldsymbol{\theta}$ and an input $\mathcal{G}_v^{sub}$, an embedding vector $\mathbf{e}_v$ is a sufficient subgraph embedding for $\mathcal{G}_v^{sub}$ if optimizing $\boldsymbol{\theta}$ with $\mathcal{G}_v^{sub}$ or $\mathbf{e}_v$ are equivalent.*

Given the definition, we aim to derive SSEs from the computation subgraphs. As we have shown in Section 3.2, SSEs cannot be derived from the MPNNs due to their interleaved neighborhood aggregation and feature transformations, *i.e.*, whenever the trainable parameters get updated, recalculating the representation of $v$ requires all nodes and edges of $\mathcal{G}_v^{sub}$. To resolve this issue, we formulate the Parameter Decoupled Graph Neural Networks (PDGNNs) framework, which decouples the trainable parameters from the individual nodes/edges. PDGNNs may not be the only feasible framework to derive SSEs, but is the first attempt in this direction and is empirically verified to be effective. Given a computation subgraph $\mathcal{G}_v^{sub}$, the prediction of node $v$ with PDGNNs consists of two steps. First, the topological information of $\mathcal{G}_v^{sub}$ is encoded into an embedding $\mathbf{e}_v$ via the function $\mathrm{f}_{topo}(\cdot)$ without trainable parameters (instantiations of $\mathrm{f}_{topo}(\cdot)$ are detailed in Section 3.4).

$$\mathbf{e}_v = \mathrm{f}_{topo}(\mathcal{G}_v^{sub}). \tag{4}$$

Next, $\mathbf{e}_v$ is further passed into a trainable function $\mathrm{f}_{out}(\cdot; \boldsymbol{\theta})$ parametrized by $\boldsymbol{\theta}$ (instantiations of $\mathrm{f}_{out}(\cdot; \boldsymbol{\theta})$ are detailed in Section 3.4) to get the output prediction $\hat{\mathbf{y}}_v$,

$$\hat{\mathbf{y}}_v = \mathrm{f}_{out}(\mathbf{e}_v; \boldsymbol{\theta}). \tag{5}$$

With the formulations above, $\mathbf{e}_v$ derived in Eq. (4) clearly satisfies the requirements of SSE (Definition 1). Specifically, since the trainable parameters acts on $\mathbf{e}_v$ instead of directly on any individual node/edge, optimizing the model parameters $\boldsymbol{\theta}$ with either $\mathbf{e}_v$ or $\mathcal{G}_v^{sub}$ are equivalent.

Since SSEs are equivalent to the computation subgraphs for optimizing PDGNNs, the memory buffer only needs to store SSEs to reduce the space complexity from $\mathcal{O}(nd^L)$ to $\mathcal{O}(n)$. For convenience, we refer to $\mathcal{G}_v^{sub}$ as the computation subgraph of both $v$ and $\mathbf{e}_v$. We name the buffer to store the SSEs as Sufficient Subgraph Embedding Memory ($\mathcal{SSEM}$). Given a new task $\tau$, the update of $\mathcal{SSEM}$ is:

$$\mathcal{SSEM} = \mathcal{SSEM} \bigcup \mathrm{sampler}(\{\mathbf{e}_v \mid v \in \mathbb{V}_\tau\}, n), \tag{6}$$

where $\mathrm{sampler}(\cdot, \cdot)$ denotes the adopted sampling strategy to populate the buffer, $\bigcup$ denotes the set union, and $n$ is the budget size. As long as a memory buffer $\mathcal{SSEM}$ is maintained, our PDGNNs-SSEM perform well with different sampling strategies including random sampling. But in Section 3.6, based on the theoretical insights in Section 3.5, we propose a special sampling strategy to better populate $\mathcal{SSEM}$, which is empirically verified to be beneficial when the memory budget is tight. Equation (6) assumes a scenario where all data of the current task are presented concurrently. In practice, if the data of a task are presented in multiple batches (*e.g.*, nodes come in batches on large graphs), the buffer update can be modified by adopting mechanisms to replace the existing data, which is detailed in Appendix A. For task $\tau$ with graph $\mathcal{G}_\tau$, the loss with $\mathcal{SSEM}$ then becomes:

$$\mathcal{L} = \underbrace{\sum_{v \in \mathbb{V}_\tau} l(\mathrm{f}_{out}(\mathbf{e}_v; \boldsymbol{\theta}), \mathbf{y}_v)}_{\mathcal{L}_\tau: \text{ loss of the current task } \tau} + \lambda \underbrace{\sum_{\mathbf{e}_w \in \mathcal{SSEM}} l(\mathrm{f}_{out}(\mathbf{e}_w; \boldsymbol{\theta}), \mathbf{y}_w)}_{\mathcal{L}_{aux}: \text{ auxiliary loss}}, \tag{7}$$

where the $\mathbf{e}_v$ on the current task is calculated according to Equation (8). Different from traditional continual learning works which choose $\lambda$ manually, on graph data, we re-scale the losses according to the class sizes to counter the bias from the severe class imbalance, which cannot be handled on graphs by directly balancing the datasets (details are provided in Appendix C.2).

### 3.4 INSTANTIATIONS OF PDGNNs

Although without trainable parameters, the function $\mathrm{f}_{topo}(\cdot)$ for generating SSEs can be highly expressive with various formulations including linear and non-linear ones, both of which are studied in this work. We will mainly focus on the linear formulations, which are empirically comparable to the non-linear choices (Appendix C.3) but is much more efficient and convenient for theoretical analysis (Section 3.5 and 3.6). The linear instantiations of $\mathrm{f}_{topo}(\cdot)$ can be generally formulated as,

$$\mathbf{e}_v = \mathrm{f}_{topo}(\mathcal{G}_v^{sub}) = \sum_{w \in \mathbb{V}} \mathbf{x}_w \cdot \pi(v, w; \hat{\mathbf{A}}), \tag{8}$$

where $\pi(\cdot, \cdot; \hat{\mathbf{A}})$ denotes the adopted strategy for computation subgraph construction based on the structure $\hat{\mathbf{A}}$ (the normalized adjacency matrix defined in Section 3.1).

Next, to instantiate $\pi(\cdot, \cdot; \hat{\mathbf{A}})$, we first formulate the SSE generation for all nodes in $\mathbb{V}$ as a matrix multiplication: $\mathbf{E}_{\mathbb{V}} = \mathbf{\Pi} \mathbf{X}_{\mathbb{V}}$, where each entry $\mathbf{\Pi}_{v,w} = \pi(v, w; \hat{\mathbf{A}})$. $\mathbf{E}_{\mathbb{V}} \in \mathbb{R}^{|\mathbb{V}| \times b}$ is the concatenation of all SSEs ($\mathbf{e}_v \in \mathbb{R}^b$), and $\mathbf{X}_{\mathbb{V}} \in \mathbb{R}^{|\mathbb{V}| \times b}$ is the concatenation of all node feature vectors $\mathbf{x}_v \in \mathbb{R}^b$. The following three options are adopted as instantiations of $\mathbf{\Pi}$ in our experiments:

1. SGC Wu et al. (2019a): $\mathbf{\Pi} = \hat{\mathbf{A}}^L$

2. S$^2$GC Zhu & Koniusz (2020): $\mathbf{\Pi} = \frac{1}{L} \sum_{l=1}^{L} \left( (1 - \alpha) \hat{\mathbf{A}}^l + \alpha \mathbf{I} \right)$

3. APPNP Klicpera et al. (2018): $\mathbf{\Pi} = \left((1 - \alpha)\hat{\mathbf{A}} + \alpha\mathbf{I}\right)^L$

Note that PDGNNs is a general framework, and some existing decoupled GNNs are instantiations of PDGNNs, *e.g.,* SGC Wu et al. (2019a) and S$^2$GC Zhu & Koniusz (2020). However, the other decoupled GNNs may not decouple the trainable parameters from individual nodes, and the space complexity is still $\mathcal{O}(nd^L)$ when applying memory replay, *e.g.,* APPNP Klicpera et al. (2018), PTA Dong et al. (2021), *etc.*.

The linear formulation of $\mathrm{f}_{topo}(\cdot)$ described above in Equation 8 could yield both promising experimental results (Section C) and instructive theoretical results (Section 3.5 and 3.6). Equation 8 is also highly efficient especially for large graphs due to the absence of iterative neighborhood aggregations. Besides, $\mathrm{f}_{topo}(\cdot)$ can also take non-linear forms with more complex mappings, *e.g.,* the reservoir computing modules Gallicchio & Micheli (2020). The corresponding experimental and theoretical effects are introduced in Appendix (B.3 and C.5).

Since the function $\mathrm{f}_{out}(\cdot; \boldsymbol{\theta})$ simply deals with individual vectors (SSEs), it is instantiated as MLP in this work. The specific configurations of $\mathrm{f}_{out}(\cdot; \boldsymbol{\theta})$ is described in the experimental part (Section 4.2).

## 3.5 PSEUDO-TRAINING EFFECTS OF SSEs

In traditional continual learning on Euclidean data without topological connections, replaying an example $\mathbf{x}_i$ (*e.g.,* an image) only reinforces the prediction of $\mathbf{x}_i$ itself. In this subsection, we introduce the pseudo-training effect of SSEs, which implies that training PDGNNs with $\mathbf{e}_v$ of node $v$ also influences the predictions of the other nodes in $\mathcal{G}_v^{sub}$, based on which we develop a novel sampling strategy to further boost the continual learning performance on graphs.

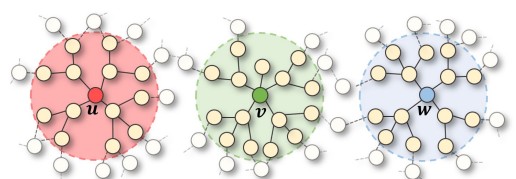

Figure 2: Illustration of the coverage ratio. Supposing the graph has $N$ nodes in total, $R_c(\{u\}) = \frac{13}{N}$, $R_c(\{v\}) = \frac{15}{N}$, $R_c(\{u\}) = \frac{14}{N}$, and $R_c(\{u, v, w\}) = \frac{42}{N}$

**Theorem 1** (Pseudo-training). *Given a node $v$, its computation subgraph $\mathcal{G}_v^{sub}$, the SSE $\mathbf{e}_v$, and label $\mathbf{y}_v$ (suppose $v$ belongs to class $k$, i.e. $\mathbf{y}_{v,k} = 1$), then training PDGNNs with $\mathbf{e}_v$ has the following two properties:*

*1. It is equivalent to training PDGNNs with each node $w$ in $\mathcal{G}_v^{sub}$ with $\mathcal{G}_v^{sub}$ being a pseudo computation subgraph and $\mathbf{y}_v$ being a pseudo label, where the contribution of $\mathbf{x}_w$ (via Equation 8) is re-scaled by $\frac{\pi(v,w;\hat{\mathbf{A}})}{\pi(w,w;\hat{\mathbf{A}})}$. We term this property as the pseudo-training effect on neighboring nodes, because it is equivalent to that the training is conducted on each neighboring node (in $\mathcal{G}_v^{sub}$) through the pseudo labels and the pseudo computation subgraphs.*

*2. When $\mathrm{f}_{out}(\cdot; \boldsymbol{\theta})$ is linear, training PDGNNs on $\mathbf{e}_v$ is also equivalent to training $\mathrm{f}_{out}(\cdot; \boldsymbol{\theta})$ on pseudo-labeled nodes $(\mathbf{x}_w, \mathbf{y}_v)$ for each $w$ in $\mathcal{G}_v^{sub}$, where the contribution of $w$ in the loss is adaptively re-scaled with a weight $\frac{\mathrm{f}_{out}(\mathbf{x}_w; \boldsymbol{\theta})_k \cdot \pi(v,w;\hat{\mathbf{A}})}{\sum_{w \in \mathbb{V}_v^{sub}} \mathrm{f}_{out}\left(\mathbf{x}_w \cdot \pi(v,w;\hat{\mathbf{A}}); \boldsymbol{\theta}\right)_k}$.*

The pseudo-training effect essentially arises from the neighborhood aggregation operation. Due to the prevalence of homophily (defined in Appendix B.2) in real-world graphs, neighborhood aggregation (*i.e.,* message passing) is widely adopted in mainstream GNNs to enhance the performance by encouraging similar representations and predictions for neighbored nodes. Similarly, pseudo-training effect implies that replaying the SSE of a buffered node is encouraging a similar prediction for its neighbors (not buffered), which is also beneficial on homophilous graphs. In other words, the homophilous neighbors of a buffered node $v$ do not need to be stored, but the forgetting problem on them can also be alleviated by replaying the SSE of $v$. Besides, when $\mathrm{f}_{out}(\cdot; \boldsymbol{\theta})$ is linear, the re-scaling weight in Theorem 1.2 can adjust the pseudo-training on neighboring nodes according to their homophily to some extent. Specifically, larger $\mathrm{f}_{out}(\mathbf{x}_w; \boldsymbol{\theta})_k$ denotes a higher confidence to classify $w$ into class $k$, and a higher $\pi(v, w; \hat{\mathbf{A}})$ typically denotes a higher similarity between $w$ and $v$. Therefore, the pseudo-training is stronger on the homophilous neighbors (with same labels) and weaker on the heterophilous neighbors (with different labels), according to the prediction confidence of $\mathrm{f}_{out}(\cdot; \boldsymbol{\theta})$. Note that although homophily brings this extra benefit, it is not a prerequisite for our model to work. Despite the homophily, replaying $\mathbf{e}_v$ still reinforces the prediction of node $v$ itself, just like memory replay in traditional continual learning on independent data (e.g., images). However, real-world graphs often exhibit strong homophily, and pseudo-training effect is generally beneficial, which is also empirically justified (Section 4.3).

Table 1: The detailed statistics of datasets and task splittings

| Dataset | CoraFull McCallum et al. (2000) | OGB-Arxiv[1] | Reddit Hamilton et al. (2017) | OGB-Products[2] |
|---|---|---|---|---|
| # nodes | 19,793 | 169,343 | 232,965 | 2,449,029 |
| # edges | 130,622 | 1,166,243 | 114,615,892 | 61,859,140 |
| # classes | 70 | 40 | 40 | 47 |
| # tasks | 30 / 14 / 5 / 2 | 20 / 8 / 5 / 2 | 20 / 8 / 5 / 2 | 23 / 10 / 5 / 3 |

The above analysis suggests that SSEs with larger computation graphs covering more nodes may be more effective. In the next subsection, we design the coverage maximization sampling strategy to leverage the benefit of the pseudo-training effect.

### 3.6 COVERAGE MAXIMIZATION SAMPLING

Following the above subsection, to quantify the number of nodes covered by the selected SSEs versus the total number of nodes in the graph, we define the coverage ratio of the SSEs. In the following, since each SSE uniquely corresponds to a node, we may use 'node' and 'SSE' interchangeably

**Definition 2.** *Given a graph $\mathcal{G}$, node set $\mathbb{V}$, and function $\pi(\cdot, \cdot; \hat{\mathbf{A}})$, the coverage ratio of a set of nodes $\mathbb{V}_s$ is:*

$$R_c(\mathbb{V}_s) = \frac{|\cup_{v \in \mathbb{V}_s} \{w | w \in \mathcal{G}_v^{sub}\}|}{|\mathbb{V}|}, \quad (9)$$

*i.e., the ratio of nodes of the entire (training) graph covered by the computation subgraphs of the selected nodes (SSEs).*

---

**Algorithm 1** Coverage maximization sampling

**Input:** $\mathcal{G}_\tau, \mathbb{V}_\tau, \hat{\mathbf{A}}_\tau, \pi(\cdot, \cdot; \cdot)$, sample size $n$.
**Output:** Selected nodes $\mathcal{S}$
1: Initialize $\mathcal{S} = \{\}$.
2: **for each** $v \in \mathbb{V}_\tau$ **do**
3:     $R_c(\{v\}) = \frac{|\{w | w \in \mathcal{G}_{\tau,v}^{sub}\}|}{|\mathbb{V}_\tau|}$
4: **end for each**
5: **for each** $v \in \mathbb{V}_\tau$ **do**
6:     $p_v = \frac{R_c(\{v\})}{\sum_{w \in \mathbb{V}_\tau} R_c(\{w\})}$
7: **end for each**
8: **while** $n > 0$ **do**
9:     Sample one node $v$ from $\mathbb{V}_\tau$ according to $\{p_w \mid w \in \mathbb{V}_\tau\}$.
10:     $\mathcal{S} = \mathcal{S} \cup \{v\}$
11:     $\mathbb{V}_\tau = \mathbb{V}_\tau \backslash \{v\}$ ▷ Sampling without replacement
12:     $n \leftarrow n - 1$
13: **end while**

---

To maximize $R_c(\mathcal{SSEM})$, a naive approach is to start from the SSE with the largest coverage ratio and iteratively incorporate SSE that increases $R_c(\mathcal{SSEM})$ the most. However, this requires computing $R_c(\mathcal{SSEM})$ for all possible SSEs in each iteration, which is time consuming especially on large graphs. Besides, certain randomness is also desired for the diversity of $\mathcal{SSEM}$. Therefore, we propose to sample SSEs from a multinomial distribution based on the coverage ratio of each individual SSE. Specifically, in task $\tau$ with node set $\mathbb{V}_\tau$, the probability of sampling node $v \in \mathbb{V}_\tau$ is $p_v = \frac{R_c(\{v\})}{\sum_{w \in \mathbb{V}_\tau} R_c(\{w\})}$. Then the procedure is to sample from $\mathbb{V}_\tau$ according to $\{p_v \mid v \in \mathbb{V}_\tau\}$ without replacement, as shown in Algorithm 1. In experiments, we compare different sampling strategies to demonstrate the strong correlation between the coverage ratio and the performance, which also verifies the benefits revealed in Section 3.5

### 4 EXPERIMENTS

In this section, we aim to answer the following questions: Q1: Whether PDGNNs-SSEM work well with a reasonable buffer size? Q2: Does coverage maximization sampling ensure a higher coverage ratio? Q3: Whether our theoretical results can be empirically justified? Q4: Does a higher coverage ratio lead to better performance? Q5: Whether PDGNNs-SSEM can outperform the state-of-the-art methods in both class-IL and task-IL scenarios? Due to the space limitations, only the most prominent results are presented in the main content, and more details are available in Appendix. For simplicity, PDGNNs-SSEM will be denoted as PDGNNs in this section.

### 4.1 DATASETS

We adopted four public datasets, CoraFull, OGB-Arxiv, Reddit, and OGB-Products, with up to millions of nodes and 70 classes. Dataset statistics and task splittings (*i.e.*, how we partition the node classes into different tasks) are summarized in Table 5. In the paper, we show the results under the splittings with the largest number of tasks. More details of the datasets, dataset splittings, and the results with other task splittings are provided in the Appendix C.1,C.2,C.4.

### 4.2 EXPERIMENTAL SETUP AND MODEL EVALUATION

**Continual learning setting and model evaluation.** During training, a model is trained on a task sequence with access only to the subgraphs of the current task. After that, the model is tested on all learned tasks. In the class-IL scenario, a model has to classify a given node by picking a class from all learned classes (which is more challenging), while the task-IL scenario only requires the model

---

[1] https://ogb.stanford.edu/docs/nodeprop/#ogbn-arxiv
[2] https://ogb.stanford.edu/docs/nodeprop/#ogbn-products

Table 2: Performance & coverage ratios of different sampling strategies and buffer sizes on OGB-Arxiv dataset (↑ higher means better).

| | Ratio of dataset /% | 0.02 | 0.1 | 1.0 | 5.0 | 40.0 |
|---|---|---|---|---|---|---|
| AA % | Uniform samp. | 12.0±1.1 | 24.1±1.7 | 42.2±0.3 | 50.4±0.4 | 53.3±0.4 |
| | Mean of feat. | 12.6±0.1 | 25.3±0.3 | 42.8±0.3 | 50.4±0.7 | 53.3±0.2 |
| | Cov. Max. | **14.9±0.8** | **26.8±1.8** | **43.7±0.5** | **50.5±0.4** | **53.4±0.1** |
| Cov. ratio/% | Uniform samp. | 0.1±0.1 | 0.3±0.0 | 3.5±0.9 | 15.9±1.1 | 84.8±1.5 |
| | Mean of feat. | 0.2±0.4 | 0.6±0.3 | 7.1±0.6 | 29.6±1.7 | 91.1±0.1 |
| | Cov. Max. | **0.5±1.1** | **2.9±1.8** | **22.5±1.6** | **46.3±0.6** | **92.8±0.0** |

Table 3: Performance comparisons under class-IL on different datasets (↑ higher means better).

| C.L.T. | CoraFull | | OGB-Arxiv | | Reddit | | OGB-Products | |
|---|---|---|---|---|---|---|---|---|
| | AA/% ↑ | AF/% ↑ | AA/% ↑ | AF /% ↑ | AA/% ↑ | AF /% ↑ | AA/% ↑ | AF /% ↑ |
| Fine-tune | 3.5±0.2 | -95.2±0.3 | 4.9±0.0 | -89.7±0.4 | 5.9±0.9 | -97.9±1.8 | 7.6±0.6 | -88.7±0.5 |
| EWC 2017 | 52.6±5.0 | -38.5±7.3 | 8.5±0.6 | -69.5±4.7 | 10.3±6.3 | -33.2±14.6 | 23.8±2.2 | -21.7±3.9 |
| MAS 2018 | 6.5±1.0 | -92.3±1.0 | 4.8±0.2 | -72.2±2.6 | 9.2±7.6 | -23.1±14.6 | 16.7±2.6 | -57.0±24.8 |
| GEM 2017 | 7.4±0.1 | -91.0±0.1 | 4.9±0.0 | -89.8±0.2 | 5.0±0.0 | -99.4±0.0 | 4.5±0.8 | -94.7±0.2 |
| TWP 2021 | 62.6±1.7 | -30.6±2.8 | 6.7±1.2 | -50.6±6.9 | 8.0±2.9 | -18.8±5.1 | 14.1±2.1 | -11.4±1.3 |
| LwF 2017 | 33.4±0.9 | -59.6±1.2 | 9.9±6.7 | -43.6±7.5 | 86.6±0.8 | -9.2±0.9 | 48.2±0.8 | -18.6±0.9 |
| ER-GNN 2021 | 2.9±0.0 | -94.6±0.1 | 12.3±3.1 | -79.9±3.3 | 20.4±2.6 | -82.7±2.9 | 56.7±0.3 | -33.3±0.5 |
| Joint | 80.8±0.1 | -3.1±0.2 | 56.8±0.0 | -8.6±0.0 | 97.1±0.1 | -0.7±0.1 | 71.5±0.1 | -5.8±0.2 |
| **PDGNNs** | **81.9±0.1** | **-3.9±0.1** | **53.2±0.2** | **-14.7±0.2** | **96.6±0.0** | **-2.6±0.1** | **73.9±0.1** | **-10.9±0.2** |

to distinguish the classes within each task. For model evaluation, the most thorough metric is the accuracy matrix $M^{acc} \in \mathbb{R}^{T \times T}$, where $M_{i,j}^{acc}$ denotes the accuracy on task $j$ after learning task $i$. The learning dynamics are shown with the curves of average accuracy (AA): $\left\{ \frac{\sum_{j=1}^{i} M_{i,j}^{acc}}{i} | i = 1, ..., T \right\}$ and the average forgetting (AF): $\left\{ \frac{\sum_{j=1}^{i-1} M_{i,j}^{acc} - M_{j,j}^{acc}}{i-1} | i = 2, ..., T \right\}$ when the number of learned tasks varies. To use a single numeric value for evaluation, the AA and AF after learning all $T$ tasks will be used. We repeat all experiments 5 times on one Nvidia Titan Xp GPU. All results are reported with average performance and standard deviations.

**Baselines and model settings.** Our baselines include the methods designed for continual graph learning including Experience Replay based GNN (ERGNN) (Zhou & Cao, 2021) and Topology-aware Weight Preserving (TWP) (Liu et al., 2021), and milestone works designed for Euclidean data but also applicable to GNNs including Elastic Weight Consolidation (EWC) (Kirkpatrick et al., 2017), Learning without Forgetting (LwF) (Li & Hoiem, 2017), Gradient Episodic Memory (GEM) (Lopez-Paz & Ranzato, 2017), and Memory Aware Synapses (MAS) (Aljundi et al., 2018)). These baselines are implemented based on three popular backbone GNNs, *i.e.*, Graph Convolutional Networks (GCNs) (Kipf & Welling, 2016), Graph Attentional Networks (GATs) (Veličković et al., 2017), and Graph Isomorphism Network (GIN) (Xu et al., 2018). Besides, joint training (without forgetting problem) and fine-tune (without continual learning technique) are adopted as the upper and lower bound for performance comparison. We instantiate $f_{out}(\cdot; \boldsymbol{\theta})$ as a multi-layer perceptron (MLP). To make a fair comparison, all methods including $f_{out}(\cdot; \boldsymbol{\theta})$ of PDGNNs are set as 2-layer with 256 hidden dimensions, and the neighborhood aggregation range of PDGNNs ($L$ in Section 3.3) is also set as 2 for consistency. As detailed in Section 4.3, $f_{topo}(\cdot)$ is chosen as the SGC strategy (Section 3.4), while the comparison among different choices is introduced in Appendix C.4.

### 4.3 STUDIES ON THE BUFFER SIZE & PERFORMANCE VS. COVERAGE RATIO (Q1,2,3,4)

In Table 2, based on PDGNNs, we compare the proposed *coverage maximization sampling* with uniform sampling and mean of feature (MoF) in terms of coverage ratios and performance when the buffer size (ratio of dataset) varies from 0.0002 to 0.4 on OGB-Arxiv. More complicated sampling methods Hübler et al. (2008); Yang et al. (2016) can also be used. However, the sampling methods adopted in this work can already obtain performance comparable to Joint, and are highly efficient. The proposed *coverage maximization sampling* achieves a superior coverage ratio, especially when buffer sizes are relatively small. We also notice that the average accuracy for *coverage maximization sampling* is positively related to the coverage ratio in general, which verifies Theorem 1.
Table 2 also shows the positive correlation between the buffer size and the performance. Besides, our SSEM appears to be highly efficient in terms of memory usage. No matter which sampling strategy is used, the performance can reach ≈50 average accuracy (AA) with only 5% data buffered. In Appendix C.5, we further evaluate how the performance changes when the buffer size varies with different variants of PDGNNs (*i.e.*, the SSE generation strategies adopted from SGC, S²GC, APPNP,

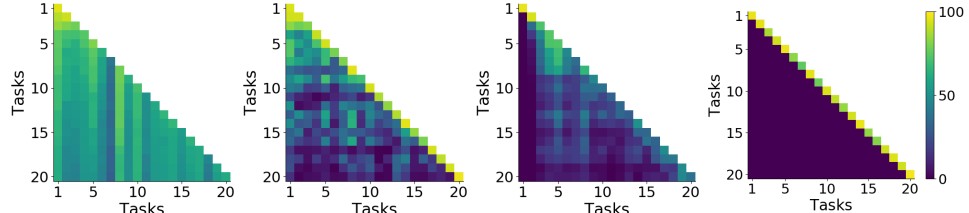

Figure 3: Dynamics of average accuracy in class-IL scenario. From left to right: 1. OGB-Arxiv, 2 classes per task. 2. CoraFull, 2 classes per task. 3. Reddit, 2 classes per task. 4. OGB-Products, 2 classes per task.

Figure 4: From left to right: accuracy matrix of PDGNNs, ER-GNN, LwF, and Fine-tune on OGB-Arxiv dataset.

Table 4: Performance comparisons under task-IL on different datasets (↑ higher means better).

| C.L.T. | CoraFull | | OGB-Arxiv | | Reddit | | OGB-Products | |
|---|---|---|---|---|---|---|---|---|
| | AA/% ↑ | AF/% ↑ | AA/% ↑ | AF /% ↑ | AA/% ↑ | AF /% ↑ | AA/% ↑ | AF /% ↑ |
| Fine-tune | 56.0±2.4 | -41.0±2.5 | 56.2±1.7 | -36.2±1.7 | 79.5±13.0 | -11.7±2.9 | 64.4±2.3 | -31.1±2.6 |
| EWC 2017 | 89.8±0.7 | -5.1±0.3 | 71.5±0.4 | -0.9±0.4 | 83.9±11.9 | -2.0±1.0 | 87.0±0.9 | -1.7±0.9 |
| MAS 2018 | 92.2±0.6 | -3.7±0.8 | 72.7±1.6 | -18.5±1.6 | 61.1±4.3 | -0.5±0.7 | 80.6±2.4 | -13.7±2.4 |
| GEM 2017 | 92.0±0.4 | 0.3±0.8 | 80.8±0.8 | -5.3±0.9 | **98.9±0.0** | **-0.5±0.1** | 87.7±1.1 | -7.0±1.2 |
| TWP 2021 | 94.3±0.5 | -1.6±0.3 | 80.9±1.0 | -1.3±0.8 | 78.0±13.4 | -0.2±0.3 | 81.8±2.2 | -0.3±0.5 |
| LwF 2017 | 93.8±0.1 | -0.4±0.1 | 71.1±1.7 | -1.5±0.5 | 98.6±0.1 | -0.0±0.0 | 86.3±0.1 | -0.5±0.1 |
| ER-GNN 2021 | 62.4±1.5 | -34.5±1.5 | 86.4±0.2 | 0.5±0.3 | 97.5±1.5 | 2.6±3.7 | 86.4±0.0 | 11.7±0.0 |
| Joint | 96.0±0.1 | 0.0±0.1 | 90.3±0.2 | 0.5±0.2 | 99.5±0.0 | 0.0±0.0 | 95.3±0.4 | -0.3±0.3 |
| **PDGNNs** | **94.6±0.1** | **0.6±1.0** | **89.8±0.4** | **-0.0±0.5** | **98.9±0.0** | **-0.5±0.0** | **93.5±0.5** | **-2.1±0.1** |

and reservoir computing described in Section 3.3). The SGC strategy is more efficient than the other variants with comparable performance, therefore is chosen in our following experiments.

## 4.4 RESULTS FOR CLASS-IL SCENARIO AND TASK-IL SCENARIO (Q5)

**Class-IL Scenario**. We compare PDGNNs with the baselines on 4 public datasets under the class-IL scenario. As shown in Table 3, PDGNNs significantly outperform the baselines and is even comparable to joint training (the performance upper bound) on 4 different datasets. The learning dynamics are also shown in Figure 3. Among the baselines, those techniques relying on regularization or Fine-tune exhibit severe forgetting problems. LwF performs slightly better than them since knowledge distillation is employed. ER-GNN outperforms LwF since it leverages memory replay to maintain performance over old tasks. For clarity, we omit the error bars on the CoraFull dataset. Full results with error bars are available in Appendix C.4.

To further understand the dynamics of different methods under the class-IL scenario, we visualize the accuracy matrices of PDGNNs, ER-GNN, LwF, and Fine-tune in Figure 4. Each row of the matrix denotes the performance on all tasks after learning a new task, and each column denotes the performance change of a specific task when learning all tasks sequentially. Compared to baselines exhibiting severer forgetting when learning new tasks, PDGNNs can maintain relatively stable performance on each task even though new tasks are continuously learned. Besides, we also visualized the learnt node representations of after learning all tasks, which is shown in Appendix C.4.

**Task-IL Scenario.** In Table 4, we can observe that PDGNNs still outperform baselines on all 4 different datasets under the task-IL scenario even though it is less challenging than the class-IL scenario as we discussed in Section 4.2. Due to space limitations, more detailed discussions about the results and the learning dynamics with the task-IL scenario are provided in Appendix C.4.

## 5 CONCLUSION

In this work, we propose the PDGNNs with SSEM for continual graph representation learning. Based on SSEs, we reduce the memory space complexity from $\mathcal{O}(nd^L)$ to $\mathcal{O}(n)$, which enables PDGNNs to fully utilize the explicit topological information sampled from previous tasks. We also discover and theoretically analyze the pseudo-training effect of SSEs. This inspires us to develop *coverage maximization sampling* which has been demonstrated to be highly efficient especially when the memory budget is tight. Finally, thorough empirical studies on both class-IL and task-IL continual learning scenarios demonstrate the effectiveness of PDGNNs-SSEM.

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

# A  ADDITIONAL DETAILS ON PARAMETER DECOUPLED GNNS WITH SSEM

As mentioned in Section 3.3 of the paper, in real-world applications, the data may come in batches instead of being presented simultaneously. Therefore, the updating of $\mathcal{SSEM}$ may need modification. The key issue is to determine how to update $\mathcal{SSEM}$ such that the newly sampled SSEs can be accommodated accordingly. We present two different approaches to handle this.

1. The most straightforward approach is to store the computation subgraph size $s_v^{sub}$ of each $\mathbf{e}_v$ and recalculate the multinomial distribution. Given the incoming new node set $\mathbb{V}_\tau$, the probability of sampling each node is recalculated as $p_v = \frac{s_v^{sub}}{\sum_{w \in \mathbb{V}_\tau \cup \mathcal{SSEM}_i} s_w^{sub}}$. Then $n$ SSEs are sampled to populate the $\mathcal{SSEM}$.

2. For efficiency, we can also adopt the reservoir sampling based strategy to update existing SSEs in $\mathcal{SSEM}$ without recalculating the multinomial distribution. Specifically, given a new node set $\mathbb{V}_\tau$, we first sample $\min\{n, |\mathbb{V}_\tau|\}$ nodes (SSEs) $\mathcal{S}$ from $\mathbb{V}_\tau$ with the coverage maximization sampling. Next, we align all SSEs in $\mathcal{SSEM}$ and $\mathcal{S}$ in a sequence, $i.e.$ the first $|\mathcal{SSEM}|$ elements are from $\mathcal{SSEM}$ and the following elements are from $\mathcal{S}$. Finally, for each SSE $\mathbf{e}_v$ in $\mathcal{S}$, suppose its order in the sequence is $o_v \in \{|\mathcal{SSEM}|, |\mathcal{SSEM}| + 1, ..., |\mathcal{SSEM}| + |\mathcal{S}|\}$, we generate a random integer $r$ from uniform distribution on 1 to $|\mathcal{SSEM}| + o_v$. If $r$ falls in the range from 1 to $|\mathcal{SSEM}|$, then the $r$-th SSEs in $\mathcal{SSEM}$ is replaced by $\mathbf{e}_v$, otherwise $\mathbf{e}_v$ is deleted. In this way, the nodes in $\mathcal{SSEM}$ can be randomly updated with the newly sampled SSEs.

# B  THEORETICAL ANALYSIS

In this section, we give proofs and detailed analysis of the theoretical results in the paper.

## B.1  PARAMETER DECOUPLED GNNS WITH SSEM

In Section 3.3, we mentioned that the embedding $\mathbf{e}_v$ derived in PDGNNs is a sufficient subgraph embedding of $\mathcal{G}_v^{sub}$ with respect to the optimization of $\boldsymbol{\theta}$. Although this is straightforward, we still provide a proof for completeness.

*Proof.* According to Definition 1, a sufficient condition for a vector $\mathbf{e}_v$ to be a sufficient subgraph embedding of $\mathcal{G}_v^{sub}$ is that $\mathbf{e}_v$ provides same information as $\mathcal{G}_v^{sub}$ for optimizing the parameter $\boldsymbol{\theta}$ of a model $f_{out}(\cdot; \boldsymbol{\theta})$. Therefore, the proof can be done by showing $\nabla_{\boldsymbol{\theta}} \mathcal{L}(\mathbf{e}_v, \boldsymbol{\theta}) = \nabla_{\boldsymbol{\theta}} \mathcal{L}(\mathcal{G}_v^{sub}, \boldsymbol{\theta})$, where $\mathcal{L}$ is the adopted loss function. This becomes straightforward under the PDGNNs framework since $\mathcal{G}_v^{sub}$ is first embedded in $\mathbf{e}_v$ and then participate in the computation with the trainable parameter $\boldsymbol{\theta}$. Specifically, given an input computation subgraph $\mathcal{G}_v^{sub}$ with the label $\mathbf{y}_v$, the corresponding prediction of PDGNNs is:

$$\hat{y}_v = f_{out}\Big( \sum_{w \in \mathbb{V}_v^{sub}} \mathbf{x}_w \cdot \pi(v, w; \hat{\mathbf{A}}); \boldsymbol{\theta} \Big), \tag{10}$$

and the loss is:

$$\mathcal{L}_v = l\left( f_{out}\Big( \sum_{w \in \mathbb{V}_v^{sub}} \mathbf{x}_w \cdot \pi(v, w; \hat{\mathbf{A}}); \boldsymbol{\theta} \Big), \mathbf{y}_v \right), \tag{11}$$

the gradient of loss $\mathcal{L}_v$ is:

$$\nabla_{\boldsymbol{\theta}} \mathcal{L}_v = \nabla_{\boldsymbol{\theta}} l\left( f_{out}\Big( \sum_{w \in \mathbb{V}_v^{sub}} \mathbf{x}_w \cdot \pi(v, w; \hat{\mathbf{A}}); \boldsymbol{\theta} \Big), \mathbf{y}_v \right). \tag{12}$$

When the input $\mathcal{G}_v^{sub}$ is replaced with $\mathbf{e}_v$, the prediction becomes:

$$\hat{\mathbf{y}}_v = f_{out}(\mathbf{e}_v; \boldsymbol{\theta}), \tag{13}$$

and the corresponding loss becomes:

$$\mathcal{L}'_v = l\big(\mathrm{f}_{out}(\mathbf{e}_v; \boldsymbol{\theta}), \mathbf{y}_v\big), \tag{14}$$

the gradient of loss $\mathcal{L}_v$ becomes:

$$\nabla_{\boldsymbol{\theta}} \mathcal{L}'_v = \nabla_{\boldsymbol{\theta}} l\big(\mathrm{f}_{out}(\mathbf{e}_v; \boldsymbol{\theta}), \mathbf{y}_v\big). \tag{15}$$

Since in the PDGNNs, $\mathbf{e}_v$ is calculated as:

$$\mathbf{e}_v = \sum_{w \in \mathbb{V}^{sub}_v} \mathbf{x}_w \cdot \pi(v, w; \hat{\mathbf{A}}), \tag{16}$$

then we have:

$$\nabla_{\boldsymbol{\theta}} \mathcal{L}_v = \nabla_{\boldsymbol{\theta}} \mathcal{L}'_v, \tag{17}$$

*i.e.*, optimizing the trainable parameters with $\mathbf{e}_v$ is equal to optimizing the trainable parameters with $\mathcal{G}^{sub}_v$. □

## B.2 PSEUDO-TRAINING EFFECTS OF SSEs

**Theorem 1** (Pseudo-training). *Given a node $v$, its computation subgraph $\mathcal{G}^{sub}_v$, the SSE $\mathbf{e}_v$, and label $\mathbf{y}_v$ (suppose $v$ belongs to class $k$, i.e. $\mathbf{y}_{v,k} = 1$), then training PDGNNs with $\mathbf{e}_v$ has the following two properties:*

*1. It is equivalent to training PDGNNs with each node $w$ in $\mathcal{G}^{sub}_v$ with $\mathcal{G}^{sub}_v$ being a pseudo computation subgraph and $\mathbf{y}_v$ being a pseudo label, where the contribution of $\mathbf{x}_w$ (via Equation 4 in the paper) is re-scaled by $\frac{\pi(v,w;\hat{\mathbf{A}})}{\pi(w,w;\hat{\mathbf{A}})}$. We term this property as the pseudo-training effect on neighboring nodes.*

*2. When $\mathrm{f}_{out}(\cdot; \boldsymbol{\theta})$ is linear, training PDGNNs on $\mathbf{e}_v$ is also equivalent to training $\mathrm{f}_{out}(\cdot; \boldsymbol{\theta})$ on pseudo-labeled nodes $(\mathbf{x}_w, \mathbf{y}_v)$ for each $w$ in $\mathcal{G}^{sub}_v$, where the contribution of $w$ in the loss is adaptively re-scaled with a weight $\frac{\mathrm{f}_{out}(\mathbf{x}_w;\boldsymbol{\theta})_k \cdot \pi(v,w;\hat{\mathbf{A}})}{\sum_{w \in \mathbb{V}^{sub}_v} \mathrm{f}_{out}\big(\mathbf{x}_w \cdot \pi(v,w;\hat{\mathbf{A}});\boldsymbol{\theta}\big)_k}$.*

*Proof of Theorem 1.1.* Theorem 1.1 is rather intuitive and easy to understand, we still provide a detailed proof for rigorousness.

Given a node $v$, the prediction is:

$$\hat{\mathbf{y}}_v = \mathrm{f}_{out}(\mathbf{e}_v; \boldsymbol{\theta}) \tag{18}$$

$\because \mathbf{e}_v = \sum_{w \in \mathbb{V}^{sub}_v} \mathbf{x}_w \cdot \pi(v, w; \hat{\mathbf{A}})$, where $\mathbb{V}^{sub}_v$ denotes the node set of the computation subgraph $\mathcal{G}^{sub}_v$, and $\hat{\mathbf{A}}$ is the adjacency matrix of $\mathcal{G}^{sub}_v$.

$\therefore$

$$\hat{\mathbf{y}}_v = \mathrm{f}_{out}\Big( \sum_{w \in \mathbb{V}^{sub}_v} \mathbf{x}_w \cdot \pi(v, w; \hat{\mathbf{A}}); \boldsymbol{\theta} \Big) \tag{19}$$

Given the target (ground truth label) of node $v$ as $y_v$, the objective function of training the model with node $v$ is formulated as:

$$\mathcal{L}_v = l\Bigg( \mathrm{f}_{out}\Big( \sum_{w \in \mathbb{V}^{sub}_v} \mathbf{x}_w \cdot \pi(v, w; \hat{\mathbf{A}}); \boldsymbol{\theta} \Big), \mathbf{y}_v \Bigg), \tag{20}$$

where $l$ could be any loss function to measure the distance between the prediction and the target.

Since $\mathbb{V}^{sub}_v$ contains both the features of node $v$ and its neighbors, Equation 20 can be further expanded to separate the contribution of node $v$ and its neighbors:

$$\mathcal{L}_v = l\Bigg( \mathrm{f}_{out}\Big( \underbrace{\mathbf{x}_v \cdot \pi(v, v; \hat{\mathbf{A}})}_{\text{information from node } v} + \underbrace{\sum_{w \in \mathbb{V}^{sub}_v \backslash \{v\}} \mathbf{x}_w \cdot \pi(v, w; \hat{\mathbf{A}})}_{\text{neighborhood information}}; \boldsymbol{\theta} \Big), \mathbf{y}_v \Bigg), \tag{21}$$

Given an arbitrary node $q \in \mathbb{V}_v^{sub}$ but $q \neq v \in \mathbb{V}_v^{sub}$ (the adjacency matrix $\hat{\mathbf{A}}$ stays the same), we can similarly obtain the loss of training the model with node $q$:

$$\mathcal{L}_q = l\left( \mathrm{f}_{out}\Big( \underbrace{\mathbf{x}_q \cdot \pi(q, q; \hat{\mathbf{A}})}_{\text{information from node } q} + \underbrace{\sum_{w \in \mathbb{V}_q^{sub} \setminus \{q\}} \mathbf{x}_w \cdot \pi(q, w; \hat{\mathbf{A}})}_{\text{neighborhood information}}; \boldsymbol{\theta} \Big), \mathbf{y}_q \right). \tag{22}$$

Since $q \in \mathbb{V}_v^{sub} \setminus \{v\}$, we rewrite Equation 21 as:

$$\mathcal{L}_v = l\left( \mathrm{f}_{out}\Big( \underbrace{\mathbf{x}_q \cdot \pi(v, q; \hat{\mathbf{A}})}_{\text{information from node } q} + \underbrace{\sum_{w \in \mathbb{V}_v^{sub} \setminus \{q\}} \mathbf{x}_w \cdot \pi(v, w; \hat{\mathbf{A}})}_{\text{neighborhood information}}; \boldsymbol{\theta} \Big), \mathbf{y}_v \right), \tag{23}$$

By comparing Equation 23 and 22, we could observe the similarity in the loss of node $v$ and $q$, and the difference lies in the contribution (weight $\pi(\cdot, \cdot; \hat{\mathbf{A}})$) of each node and the neighboring nodes ($\mathbb{V}_q^{sub}$ and $\mathbb{V}_v^{sub}$). To clearly explain the analysis in the paper that stronger homophily leads to more benefits from pseudo training effect, we give the formal definition of the graph homophily ratio. Given a graph $\mathcal{G}$, the homophily ratio is defined as the ratio of the number of edges connecting nodes with a same label and the total number of edges, i.e.

$$h(\mathcal{G}) = \frac{1}{|\mathcal{E}|} \sum_{(j,k) \in \mathcal{E}} \mathbf{1}(\mathbf{y}_j = \mathbf{y}_k), \tag{24}$$

where $\mathcal{E}$ is the edge set containing all edges, $\mathbf{y}_j$ is the label of node $j$, and $\mathbf{1}(\cdot)$ is the indicator function Ma et al. (2021). For any graph, the homophily ratio is between 0 and 1. For each computation subgraph, when the homophily ratio is high, the neighboring nodes tend to share labels with the center node, and the pseudo training would be beneficial for the performance. Many real-world graphs like the social network and citation networks tend to have high homophily ratios, and pseudo training will bring much benefit, which is shown in Section 4.3 of the paper. $\qquad \square$

*Proof of Theorem 1.2.* In this part, we choose the loss function $l$ as cross entropy $\mathrm{CE}(\cdot, \cdot)$, which is the common choice for classification problems. In the following, we will first derive the gradient of training the PDGNNs with $(\mathbf{e}_v, y_v)$. For cross entropy, we denote the one-hot vector form label as $\mathbf{y}_v$, of which the $y_v$-th element is one and other entries are zero. Given the loss of a node $v$ as shown in

the Equation 20, the gradient is derived as:

$$\nabla_{\boldsymbol{\theta}}\mathcal{L}_v = \nabla_{\boldsymbol{\theta}}\text{CE}\left(\sum_{w \in \mathbb{V}_v^{sub}} \text{f}_{out}\left(\mathbf{x}_w \cdot \pi(v, w; \hat{\mathbf{A}}); \boldsymbol{\theta}\right), \mathbf{y}_v\right) \tag{25}$$

$$= \nabla_{\boldsymbol{\theta}}\left(\mathbf{y}_{v,k} \cdot \log \sum_{w \in \mathbb{V}_v^{sub}} \text{f}_{out}\left(\mathbf{x}_w \cdot \pi(v, w; \hat{\mathbf{A}}); \boldsymbol{\theta}\right)_k\right) \tag{26}$$

$$= \mathbf{y}_{v,k} \cdot \frac{\nabla_{\boldsymbol{\theta}}\left(\sum_{w \in \mathbb{V}_v^{sub}} \text{f}_{out}\left(\mathbf{x}_w \cdot \pi(v, w; \hat{\mathbf{A}}); \boldsymbol{\theta}\right)_k\right)}{\sum_{w \in \mathbb{V}_v^{sub}} \text{f}_{out}\left(\mathbf{x}_w \cdot \pi(v, w; \hat{\mathbf{A}}); \boldsymbol{\theta}\right)_k} \tag{27}$$

$$= \mathbf{y}_{v,k} \cdot \frac{\sum_{w \in \mathbb{V}_v^{sub}} \nabla_{\boldsymbol{\theta}}\text{f}_{out}\left(\mathbf{x}_w \cdot \pi(v, w; \hat{\mathbf{A}}); \boldsymbol{\theta}\right)_k}{\sum_{w \in \mathbb{V}_v^{sub}} \text{f}_{out}\left(\mathbf{x}_w \cdot \pi(v, w; \hat{\mathbf{A}}); \boldsymbol{\theta}\right)_k} \tag{28}$$

$$= \mathbf{y}_{v,k} \cdot \frac{\sum_{w \in \mathbb{V}_v^{sub}} \nabla_{\boldsymbol{\theta}}\text{f}_{out}(\mathbf{x}_w; \boldsymbol{\theta})_k \cdot \pi(v, w; \hat{\mathbf{A}})}{\sum_{w \in \mathbb{V}_v^{sub}} \text{f}_{out}\left(\mathbf{x}_w \cdot \pi(v, w; \hat{\mathbf{A}}); \boldsymbol{\theta}\right)_k} \tag{29}$$

$$= \frac{\sum_{w \in \mathbb{V}_v^{sub}} \mathbf{y}_{v,k} \cdot \frac{\nabla_{\boldsymbol{\theta}}\text{f}_{out}(\mathbf{x}_w;\boldsymbol{\theta})_k}{\text{f}_{out}(\mathbf{x}_w;\boldsymbol{\theta})_k} \cdot \text{f}_{out}(\mathbf{x}_w; \boldsymbol{\theta})_k \cdot \pi(v, w; \hat{\mathbf{A}})}{\sum_{w \in \mathbb{V}_v^{sub}} \text{f}_{out}\left(\mathbf{x}_w \cdot \pi(v, w; \hat{\mathbf{A}}); \boldsymbol{\theta}\right)_k} \tag{30}$$

$$= \frac{\sum_{w \in \mathbb{V}_v^{sub}} \nabla_{\boldsymbol{\theta}}\text{CE}\left(\text{f}_{out}(\mathbf{x}_w; \boldsymbol{\theta}), \mathbf{y}_{v,k}\right) \cdot \text{f}_{out}(\mathbf{x}_w; \boldsymbol{\theta}) \cdot \pi(v, w; \hat{\mathbf{A}})}{\sum_{w \in \mathbb{V}_v^{sub}} \text{f}_{out}\left(\mathbf{x}_w \cdot \pi(v, w; \hat{\mathbf{A}}); \boldsymbol{\theta}\right)} \tag{31}$$

$$= \sum_{w \in \mathbb{V}_v^{sub}} \frac{\text{f}_{out}(\mathbf{x}_w; \boldsymbol{\theta}) \cdot \pi(v, w; \hat{\mathbf{A}})}{\sum_{w \in \mathbb{V}_v^{sub}} \text{f}_{out}\left(\mathbf{x}_w \cdot \pi(v, w; \hat{\mathbf{A}}); \boldsymbol{\theta}\right)} \cdot \nabla_{\boldsymbol{\theta}}\text{CE}\left(\text{f}_{out}(\mathbf{x}_w; \boldsymbol{\theta}), \mathbf{y}_v\right). \tag{32}$$

The loss of training $\text{f}_{out}(\mathbf{x}_w; \boldsymbol{\theta})$ with pairs of feature and pseudo-label $(\mathbf{x}_w, y_v)$ of all nodes of $\mathcal{G}_v^{sub}$ is:

$$\mathcal{L}_{\mathcal{G}_v^{sub}} = \sum_{w \in \mathbb{V}_v^{sub}} \text{CE}\left(\text{f}_{out}(\mathbf{x}_w; \boldsymbol{\theta}), \mathbf{y}_v\right) \tag{33}$$

$$\tag{34}$$

Then, the corresponding gradient of $\mathcal{L}_{\mathcal{G}_v^{sub}}$ is :

$$\nabla_{\boldsymbol{\theta}}\mathcal{L}_{\mathcal{G}_v^{sub}} = \sum_{w \in \mathbb{V}_v^{sub}} \nabla_{\boldsymbol{\theta}}\text{CE}\left(\text{f}_{out}(\mathbf{x}_w; \boldsymbol{\theta}), \mathbf{y}_v\right). \tag{35}$$

By comparing Equation 32 and 35, we can see that training PDGNNs with a sufficient subgraph embedding $\mathbf{e}_v$ equals to training the function $\text{f}_{out}(\cdot; \boldsymbol{\theta})$ on all nodes of the computation subgraph $\mathcal{G}_v^{sub}$ with a weight $\frac{\text{f}_{out}(\mathbf{x}_w;\boldsymbol{\theta}) \cdot \pi(v,w;\hat{\mathbf{A}})}{\sum_{w \in \mathbb{V}_v^{sub}} \text{f}_{out}\left(\mathbf{x}_w \cdot \pi(v,w;\hat{\mathbf{A}});\boldsymbol{\theta}\right)}$ on each node to rescale the contribution dynamically. $\square$

### B.3 FURTHER DISCUSSION ON PSEUDO-TRAINING EFFECTS OF GENERALIZED SSE GENERATION FUNCTION

In this subsection, we give further analysis on the pseudo training effect when the SSE generation follows the following formulation:

$$\mathbf{e}_v = \mathbf{g}(\{\mathbf{x}_w \mid w \in \mathbb{V}\}, \hat{\mathbf{A}}). \tag{36}$$

In this scenario, the pseudo training effect will depend on the specific form of $\mathbf{g}(\cdot, \cdot)$. Despite this, we can still analyze the strength of pseudo training effect with respect to the smoothness of the function and the dataset properties. First of all, the pseudo training effect exists because the GNN models generate the prediction based on a local neighborhood. Therefore, the nodes with overlapping

Table 5: The detailed statistics of datasets and task splittings

| Dataset | CoraFull McCallum et al. (2000) | OGB-Arxiv[3] | Reddit (Hamilton et al., 2017) | OGB-Products[4] |
|---|---|---|---|---|
| # nodes | 19,793 | 169,343 | 232,965 | 2,449,029 |
| # edges | 130,622 | 1,166,243 | 114,615,892 | 61,859,140 |
| # classes | 70 | 40 | 40 | 47 |
| # tasks | 30 / 14 / 5 / 2 | 20 / 8 / 5 / 2 | 20 / 8 / 5 / 2 | 23 / 10 / 5 / 3 |

neighborhood (similar inputs to the model) share similar prediction results. If their labels are shared, then training these nodes could mutually reinforce each other. Accordingly, given an arbitrary function $\mathbf{g}(\cdot, \cdot)$, we can gain an insight into the strength of pseudo training effect by analyzing the similarity of the inputs when generating representations of different nodes. Without loss of generality, we assume $\mathbf{g}(\cdot, \cdot)$ be a continuous function (since $\mathbf{g}(\cdot, \cdot)$ does not require training, it does not have to be differentiable). Then, given two nodes $v$ and $w$, we denote their corresponding inputs to the model as two vectors $I_v$ and $I_w$. $I_v$ and $I_w$ may contain different neighborhood information based on the specific form of $\mathbf{g}(\cdot, \cdot)$. Now, it is obvious that the closer $I_v$ and $I_w$ are, the closer $\mathbf{g}(I_v, \hat{\mathbf{A}})$ and $\mathbf{g}(I_w, \hat{\mathbf{A}})$ are (due to the continuity of $\mathbf{g}(\cdot, \cdot)$). In other words, stronger homophily will lead to stronger pseudo training effect as we analyzed in Theorem 1 in the paper. Besides, the frequency components (in terms of the spectrum of the function, *e.g.*, with Fourier analysis) of $\mathbf{g}(\cdot, \cdot)$ also matters. If $\mathbf{g}(\cdot, \cdot)$ is mainly composed of low frequencies, *i.e.*, the change of $\mathbf{g}(\cdot, \cdot)$ is slow with respect to the change of the input, then the pseudo training effect is stronger because more nodes are getting similar representations. But if the function $\mathbf{g}(\cdot, \cdot)$ contains strong high frequency components, *i.e.* $\mathbf{g}(\cdot, \cdot)$ changes significantly with the change of input, then the pseudo training effect is weaker since only very similar inputs of the nodes get similar outputs.

In experiments, we also instantiated $\mathbf{g}(\cdot, \cdot)$ with the reservoir computing module (Gallicchio & Micheli, 2020), which yields comparable performance with other instantiations (Section C.5).

## C    ADDITIONAL EXPERIMENTAL RESULTS

In this section, we provide additional information on the datasets, experimental settings, and experimental results.

### C.1    DATASET DESCRIPTIONS

The statistics of the datasets are summarized in Table 5. Among these datasets, CoraFull and OGB-Arxiv are two citation graphs, Reddit is a graph constructed from Reddit posts, and OGB-Products is an Amazon product co-purchasing network. The usage of the datasets is granted for academic purposes, and full details on the licenses can be obtained from the official websites. The datasets contain no personally identifiable information or offensive content.

### C.1.1    CITATION NETWORKS

CoraFull (McCallum et al., 2000) is a citation network labeled based on the paper topics. In total, it contains 19,793 nodes and 126,842 edges. The original dataset has 65,311 edges. We directly adopted the version in DGL with reverse edges added and duplicates removed. It contains 70 classes, and each node has a 8,710 dimensional feature vector.

The OGB-Arxiv dataset is collected in the Open Graph Benchmark OGB. It is a directed citation network between all Computer Science (CS) arXiv papers indexed by MAG (Wang et al., 2020d). Totally it contains 169,343 nodes and 1,166,243 edges. The dataset contains 40 classes.

---

[3]`https://ogb.stanford.edu/docs/nodeprop/#ogbn-arxiv`
[4]`https://ogb.stanford.edu/docs/nodeprop/#ogbn-products`

Table 6: The configuration of the MLP part of PDGNNs.

| No. layer | Input dimensions | Output dimensions | Activation |
|---|---|---|---|
| 1 | # data dimensions | 256 | ReLU |
| 2 | 256 | # classes | SoftMax |

### C.1.2 SOCIAL NETWORK

Reddit Hamilton et al. (2017) is a graph dataset from Reddit posts made in the month of September, 2014. The node labels are the community, or "subreddit", that the posts belong to. The authors sampled 50 large communities and built a post-to-post graph, connecting posts if the same user comments on both. In total this dataset contains 232,965 nodes with an average degree of 492, 114,615,892 edges, and a 602 dimensional feature vector for each node. We directly used the version integrated in DGL library.

### C.1.3 PRODUCT CO-PURCHASING NETWORK

OGB-Products is collected in the Open Graph Benchmark [5], representing an Amazon product co-purchasing network [6]. It contains 2,449,029 nodes and 61,859,140 edges. Nodes represent products sold on Amazon, and edges indicate that the connected products are purchased together. In our experiments, we select 46 classes and omit the last class containing only 1 example.

### C.2 ADDITIONAL DETAILS ON EXPERIMENT SETUP AND MODEL EVALUATION

**Continual learning setting**. In this part, we give concrete examples to further explain the difference between class-IL and task-IL scenarios. In class-IL scenario, a model has to classify the given data by picking a class from all previously learnt classes, while the task-IL scenario only require the model to distinguish the classes within each task. Concretely, suppose the model learns on a citation network with a two-class task sequence {(*physics*, *chemistry*), (*biology*, *math*)}. In class-IL scenario, after training, the model is required to classify a given document into one of the four classes. In task-IL scenario, the model is only required to classify a given document into to (*physics*, *chemistry*) or (*biology*, *math*), while cannot distinguish between *physics* and *biology* or between *chemistry* and *math*.

For each dataset, the splitting of different tasks is conducted by dividing the classes into groups in the default order. Different group sizes are shown in Table 1 of the paper. For each task, the ratio for training, validation, and testing is 60%, 20%, 20%. The validation set was only used in baseline model selection, since the hyperparameters of our method are simply set to be consistent with baselines (Section 4.2 in the paper).

**Baselines and model settings**. In this part, we give more details on the model configurations. The following setting applies to all datasets. All the backbone GNNs of baselines are configured as 2-layer with 256 hidden dimensions, which exhibit better performance than other configurations. To ensure a fair comparison, we also set the MLP part of PDGNNs as 2-layer with 256 hidden dimensions (the SSE generation part does not contain trainable parameters) as shown in Table 6. The memory budget (number of nodes per class selected to store) is set as 400 for PDGNNs-SSEM for all datasets. For the memory based baselines, the budget was chosen with two criteria: 1. The buffer size should be large than the size for PDGNNs-SSEM to ensure PDGNNs-SSEM does not succeed by storing more examples. 2. The budget should be large enough for the baseline methods to gain a reasonable performance. The specific budgets on different datasets are listed in Table 7, which demonstrates that PDGNNs-SEEM is actually highly efficient in using the buffered data and outperforms the memory based baselines with less memory usage. A brief introduction of the baseline continual learning techniques are given below:

---

[5]https://ogb.stanford.edu/docs/nodeprop/ogbn-products
[6]http://manikvarma.org/downloads/XC/XMLRepository.html

Table 7: Memory budget of different methods on different datasets.

|  | CoraFull | OGB-Arxiv | Reddit | OGB-Products |
|---|---|---|---|---|
| PDGNNs-SSEM | 400 | 400 | 400 | 400 |
| Baselines | 500 | 500 | 600 | 700 |

1. **Fine-tune** directly trains a given backbone GNN on the task sequence without any technique to avoid forgetting, therefore can be viewed as a lower bound on the continual learning performance.

2. **Elastic Weight Consolidation (EWC) (Kirkpatrick et al., 2017)** adds a quadratic penalty to prevent the model weights, which are important to prevent model parameters related to previous tasks from shifting too much.

3. **Memory Aware Synapses (MAS) (Aljundi et al., 2018)** measures the importance of the parameters according to the sensitivity of the predictions on the parameters and slows down the update of the important parameters.

4. **Gradient Episodic Memory (GEM) (Lopez-Paz & Ranzato, 2017)** stores representative data in episodic memory and adds a constraint to prevent the loss of the episodic memory from increasing and only allow it to decrease.

5. **Topology-aware Weight Preserving (TWP) (Liu et al., 2021)** adds a penalty on the model weights to preserve the topological information of previous graphs.

6. **Learning without Forgetting (LwF) (Li & Hoiem, 2017)** uses knowledge distillation to constrain the shift of parameters for old tasks.

7. **Experience Replay GNN (ER-GNN) (Zhou & Cao, 2021)** integrates memory-replay to GNNs by storing experience nodes from previous tasks.

8. **Joint Training** does not follow the continual learning setting and trains the model on all tasks simultaneously. Therefore, Joint Training does not suffer from forgetting problems and its performance can be viewed as the upper bound for continual learning.

A widely adopted performance upper bound on the continual learning models is joint training. Different from being trained sequentially on a task sequence, a jointly trained model does follow the continual learning setting but is simultaneously trained on all tasks. Therefore, jointly trained models do not suffer from the forgetting problem and could be viewed as an upper bound on the continual learning performance. Note that under the class-IL setting, the average accuracy of the jointly trained model will still decrease as the number of classes increases. The reason is that the classification difficulty increases when the number of classes vary from small to large.

**Class imbalance in continual graph learning**. According to Equation 37, the performance on different tasks contributes equally to the average accuracy. However, unlike the traditional continual learning with balanced datasets, the class imbalance problem is usually severe in graphs, of which the effect will be entangled with the effect of forgetting. Directly balancing the data by choosing equal number of nodes from each class may not be practical. For example, in the OGB-Products dataset, the largest class has 668,950 nodes, while the smallest contains only 1 node. Therefore, sampling equal amount of nodes from each class would result in either deleting many classes without enough nodes or sampling a very small number of nodes from each class so that all classes can provide the same amount of nodes. Moreover, deleting nodes in a graph would also change the original topological structures of the remaining nodes, which is undesired.

To this end, we propose to re-scale the loss of nodes in each class according to the class sizes. Denoting the set of the classes of our training data as $\mathcal{C}$, the number of examples of each class in $\mathcal{C}$ can be represented as $\{n_c \mid c \in \mathcal{C}\}$. Then, we calculate a scale for each class $c$ to balance their contribution in the loss function as $s_{\mathbf{y}_v} = \frac{n_c}{\sum_{i \in \mathcal{C}} n_i}$, where $\mathbf{y}_{v,c} = 1$. Finally, our balanced loss is:

$$\mathcal{L} = \sum_{v \in \mathbb{V}_\tau} l(f(\mathbf{e}_v; \boldsymbol{\theta}), \mathbf{y}_v) \cdot s_{\mathbf{y}_v} + \sum_{\mathbf{e}_w \in \mathcal{SSEM}} l(f(\mathbf{e}_w; \boldsymbol{\theta}), \mathbf{y}_w) \cdot s_{\mathbf{y}_w}. \tag{37}$$

Since the evaluation treats all classes equally and the loss on each class is balanced, $\lambda$ is omitted in our implementation, as it will influence the balance of each class.

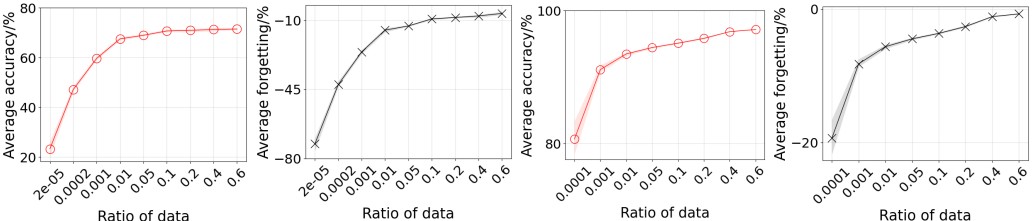

Figure 5: Average accuracy (Red circles) and average forgetting (Black crosses) changes with buffer size on OGB-Products dataset (the left two) and Reddit dataset (the right two).

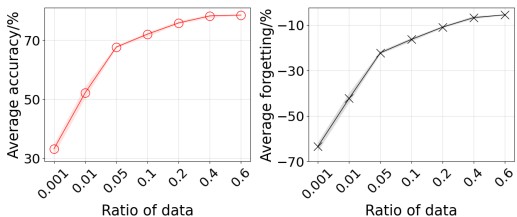

Figure 6: Average accuracy (Red circles) and average forgetting (Black crosses) changes with buffer size on CoraFull dataset.

**Class-incremental classifier**. In standard classification tasks, the number of the output heads of a model equals the number of classes and is fixed at the beginning. But in class-IL setting, the output heads will continually increase along with the new classes. To better accommodate new classes, cosine distance is adopted by several works Wu et al. (2021); Wang et al. (2018); Gidaris & Komodakis (2018) to slightly modify the standard softmax classifier. Empirically, PDGNNs with SSEM outperform the standard softmax classifier which simply increases the output heads with the number of classes. All baselines are tested with both strategies and the one that achieves better performance over the validation set is employed for comparison. Specifically, only LwF exhibits better performance with the cosine distance based classifier.

### C.3 ADDITIONAL RESULTS OF STUDIES ON THE BUFFER SIZE

In this subsection, we show the performance of PDGNNs-SSEM with different buffer sizes on the other 3 datasets in Figure 5 and 6. We observe similar patterns in these results, *i.e.*, the performance (both average accuracy and average forgetting) increases when the buffer size (in terms of the ratio of data) increases. Specifically, on OGB-Products dataset, which is the largest dataset with millions of nodes, the PDGNNs-SSEM can achieve reasonably well performance with a buffer size of only 0.01 to the size of the dataset, which further demonstrates the effectiveness and efficiency of PDGNNs-SSEM.

In Table 2 of the paper, we have the following findings: (1) our coverage maximization sampling does guarantee a superior coverage ratio compared to the other sampling strategies, especially when the buffer size is relatively small; (2) the performance does exhibit strong correlation with the coverage ratio, especially when the buffer size is small. For different buffer sizes, a higher coverage ratio can yield better performance. The performance gap between different sampling strategies is larger with smaller buffer sizes, which is also the situation when the coverage ratio gap is larger. In this case (buffer size smaller than 1.0%), the number of stored SSEs is relatively small compared to the size of the dataset, therefore the effectiveness of pseudo training on more nodes is more prominent. With larger buffer sizes, all sampling strategies can cover a large ratio of nodes and the performance gaps close up. In real world applications, a smaller buffer size is typically adopted, therefore the high memory efficiency of coverage maximization sampling would be preferred. The above analysis verifies our Theorem 1 and indicates higher coverage ratio would be beneficial to the performance.

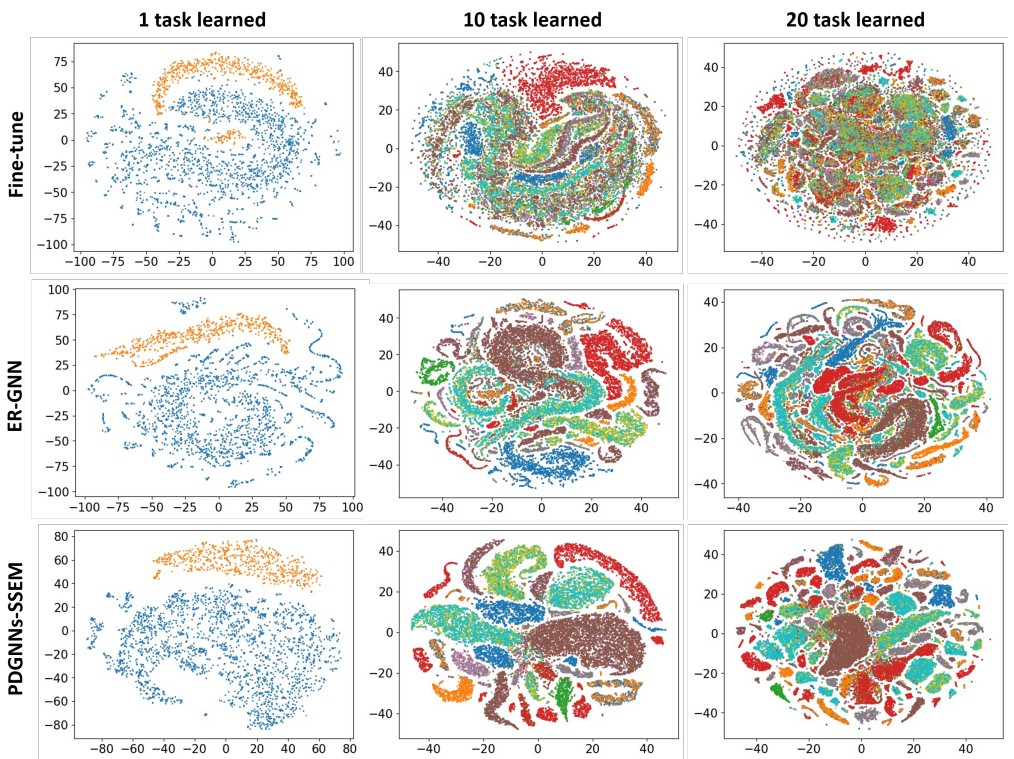

Figure 7: Visualization of node representations of different classes on Reddit dataset. The node representations are taken after learning 1, 10, and 20 tasks. From the top to the bottom, we show the results of Fine-tune, ER-GNN, and PDGNNs-SSEM. Each color corresponds to a class.

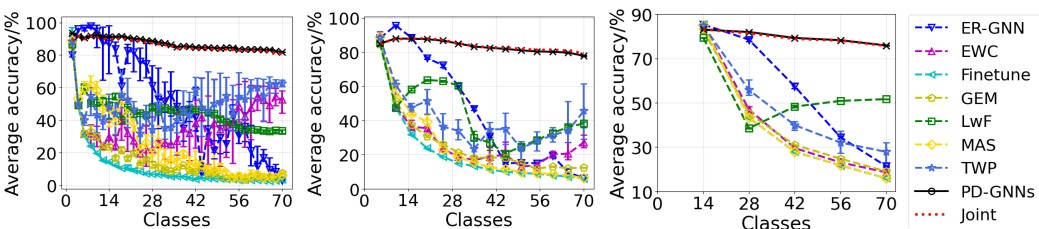

Figure 8: Dynamics of average accuracy on **CoraFull** dataset with task sequence of length of 35 (left), 14 (middle), and 5 (right) in class-IL scenario.

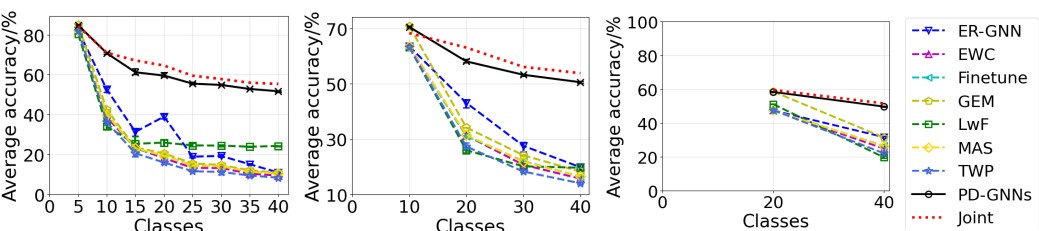

Figure 9: Dynamics of average accuracy on **OGB-Arxiv** dataset with task sequence of length of 8 (left), 4 (middle), and 2 (right) in class-IL scenario.

Table 8: Performance comparisons under class-IL on OGB-Arxiv dataset with different task splittings (↑ higher means better).

| C.L.T. | 20 tasks | | 8 tasks | | 4 tasks | | 2 tasks | |
|---|---|---|---|---|---|---|---|---|
| | AA/% ↑ | FM/%↑ | AM/% ↑ | FM /% ↑ | AM/% ↑ | FM /% ↑ | AM/% ↑ | FM /% ↑ |
| Fine-tune | 4.9±0.0 | -89.7±0.4 | 10.5±0.1 | -77.5±0.5 | 16.4±0.2 | -63.9±0.6 | 26.4±0.3 | -47.3±0.9 |
| EWC 2017 | 8.5±1.0 | -69.5±8.0 | 9.4±0.1 | -73.7±1.1 | 15.7±0.3 | -62.8±0.7 | 24.8±0.3 | -47.5±0.6 |
| MAS 2018 | 4.8±0.4 | -72.2±4.1 | 10.3±0.2 | -77.5±0.6 | 16.5±0.3 | -64.0±0.5 | 26.3±0.6 | -47.5±0.7 |
| GEM 2017 | 4.9±0.0 | -89.8±0.3 | 10.7±0.1 | -81.5±0.3 | 18.2±0.2 | -70.6±0.5 | 31.3±0.1 | -58.5±0.2 |
| TWP 2021 | 6.7±1.5 | -50.6±13.2 | 8.3±0.4 | -66.1±1.3 | 14.0±0.4 | -57.6±1.5 | 22.0±0.4 | -47.6±0.5 |
| LwF 2017 | 9.9±12.1 | -43.6±11.9 | 24.2±0.4 | -31.9±1.0 | 19.6±1.1 | -41.8±1.7 | 19.6±0.7 | -51.1±0.1 |
| ER-GNN 2021 | 12.3±3.9 | -79.9±4.1 | 10.9±0.2 | -77.5±0.5 | 19.8±1.2 | -59.9±1.3 | 31.6±0.6 | -34.8±1.3 |
| Joint | 56.8±0.0 | -8.6±0.0 | 55.3±0.0 | -10.1±0.0 | 53.9±0.0 | -9.1±0.1 | 51.6±0.1 | -8.2±0.2 |
| PDGNNs* | 26.8±1.8 | -61.6±2.0 | 27.9±1.8 | -58.2±2.7 | 30.9±1.1 | -51.2±1.6 | 35.9±1.4 | -46.4±3.2 |
| PDGNNs | 53.2±0.4 | -14.7±0.4 | 51.6±0.4 | -15.0±0.7 | 50.6±0.4 | -12.8±0.5 | 49.7±0.3 | -11.4±0.5 |

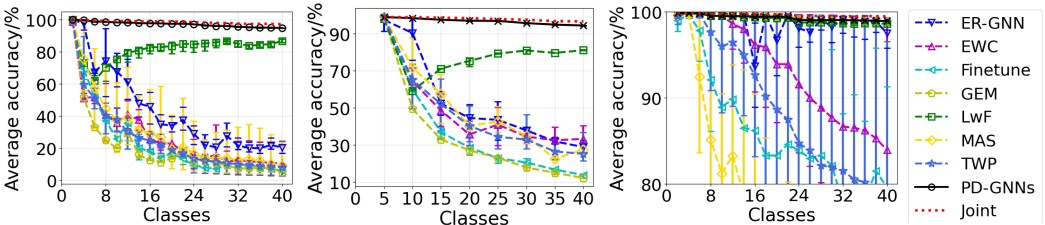

Figure 10: Dynamics of average accuracy on **Reddit** dataset with task sequence of length of 20 (left), 8 (middle) in class-IL scenario, and **Reddit** dataset with task sequence of length of 20 (right) in task-IL scenario.

## C.4    ADDITIONAL RESULTS OF COMPARISONS WITH THE STATE-OF-THE-ARTS

In this subsection, we provide additional results to compare PDGNNs-SSEM with the baselines. In Table 8, we provide numerical results to compare different models and complement the curves of average accuracy provided in the paper. We list both the final average accuracy and average forgetting of all models on the OGB-Arxiv dataset with different task splittings in class-IL scenario. Besides, we also show the results of PDGNNs-SSEM with an extremely small buffer size (*i.e.*, 0.001 of the size of the dataset), which is denoted with PDGNNs*. 0.001 of the size of OGB-Arxiv corresponds to storing only 4 examples per class and a total of 160 for 40 different classes, which is orders of magnitudes smaller than the buffer size of the memory based baselines with budgets of several hundred per class. From Table 8, we can observe that both PDGNNs and PDGNNs* significantly outperform the baselines. Even the PDGNNs* can outperform baselines by a large margin, which demonstrates the high efficiency of SSEM. Considering that OGB-Arxiv contains 169,343 nodes, the performance of PDGNNs* is indeed impressive.

Since the error bars of Figure 3 in the paper are partially omitted to highlight the performance difference of different methods, we show the complete results with error bars and the results on other

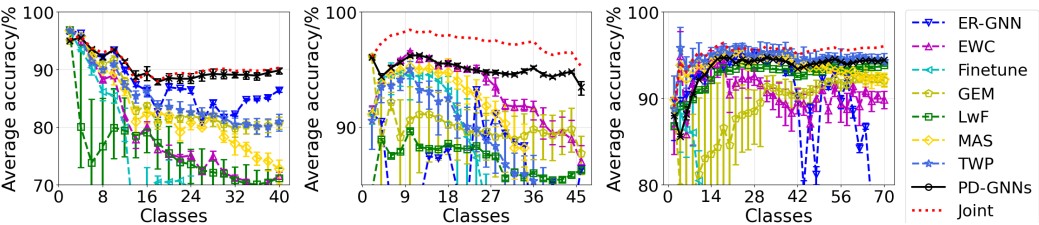

Figure 11: Dynamics of average accuracy on **OGB-Arxiv** dataset with task sequence of length of 20 (left) in task-IL scenario, **OGB-Products** dataset with task sequence of length of 23 (middle) in task-IL scenario, and **CoraFull** dataset with task sequence of length of 35 (right).

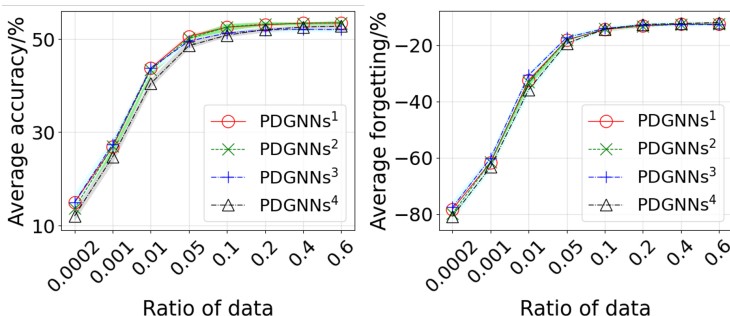

Figure 12: Average accuracy (left) and average forgetting (right) vs. buffer size on OGB-Arxiv. PDGNN[1] to PDGNN[3] instantiated $\pi(v, w; \hat{\mathbf{A}})$ as the forms introduced in Section 3.4 of the paper. PDGNN[4] adopts the reservoir computing module proposed in Gallicchio & Micheli (2020)

datasets with different task splittings (with class-IL scenario) in Figure 8, 9, and 10. Note that the task sequence of length is equivalent to the number of tasks to learn (as shown in Table 5) for each dataset.

Besides the class-IL scenario, we also provide additional results with complete error bars for the task-IL scenario in Figure 10 and 11.

To show the performance difference between PDGNNs-SSEM and the baselines more concretely, we visualize the node representations of different classes with t-SNE Van der Maaten & Hinton (2008) while learning on the task sequence (with a length of 20, *i.e.*, 20 tasks) of the Reddit dataset. In Figure 7, besides PDGNNs-SSEM, we also show two other representative baselines including ER-GNN, specially designed for continual graph learning, and Fine-tune, without continual learning techniques. According to Figure 7, PDGNNs-SSEM can maintain the nodes from different classes be well separated while continuously learning new tasks sequentially (each color corresponds to a class). In contrast, for ER-GNN and Fine-tune, the boundaries of different classes are less clear.

### C.5 ADDITIONAL STUDIES ON THE BUFFER SIZE

In Figure 12, based on the class-IL scenario, we study the performance of PDGNNs-SSEM on the OBG-Arxiv dataset when the buffer size (*i.e.*, the ratio of dataset) varies from 0.0002 to 0.6. Figure 12 exhibits the similar performance of different SSE generation modules. Besides, when the buffer size grows from 0.0002 to 0.01, both the average accuracy and average forgetting of PDGNNs increase. When the buffer size reaches 0.1, the performance of PDGNNs is comparable to the setting which stores the entire training set (when the ratio of dataset is 0.6). These results demonstrate the efficiency of SSEM. Moreover, the results in Figure 12 also show that the performance difference among different SSE generation strategies is not significant.

### D BROADER IMPACT

In this paper, we proposed a general technique to enable GNNs which can fit into the PDGNNs framework to continually learn on expanding networks. The method can be applied to any scenario requiring generating node representations on networks. The results of this paper can have an immediate and strong impact to address existing challenges for continual graph representation learning, enabling to achieve state-of-the-art performance, and thus positively impacting applications on social networks, recommender systems, dynamic systems, *etc.*

Potential negative social impact may arise depending on the application scenario. For example, the privacy issue should be carefully considered when dealing with data containing user information.

