# OpenReview forum: "Sufficient Subgraph Embedding Memory for Continual Graph Representation Learning"
_ICLR.cc/2023/Conference — Submitted to ICLR 2023_

### Official Review · Reviewer_zw6P · 2022-10-24

**Confidence:** 3
**Correctness:** 3
**Technical Novelty And Significance:** 1
**Empirical Novelty And Significance:** 2
**Recommendation:** 3

**Clarity, Quality, Novelty And Reproducibility:**

Clarity: Good

Quality: Fair

Novelty: Limited

Reproducibility: Good

**Strength And Weaknesses:**

Strengths:
1. This paper exhibits a clear logic chain to solve the memory explosion problem in the memory replay method for continual graph learning. The idea of stripping trainable parameters from the neighbor aggregation procedure is clear and effective for reducing the burden of memory replay.
2. Exhaustive experiments are conducted to compare the proposed method with baselines. The proposed PDGNNs overwhelm other baselines across all datasets and perform close to the joint-training strategy, which is carried out as the upper bound.

Weaknesses:
1. This paper is more like to integrate various existing methods. The formulation of the PDGNN with SSEM is almost identical to that of the SGC (Simplifying Graph Convolutional Networks) method. The memory replay framework and coverage maximization sampling method almost show no differences from those of the ER-GNN paper (which is also a baseline in this paper). Consequently, this paper shows limited novelty.
2. This paper claims that the memory replay benefits from the SSEM module by compressing information from subgraphs, which shows an advantage over the ER-GNN method. However, to the best of our knowledge, the ER-GNN method adopts the GAT network as its backbone, which also integrates information from neighbor nodes. Thus, the ER-GNN method can not be treated as a single-node buffer method. Consequently, the reason for the performance improvement over ER-GNN is not convincing and needs more explanation.
3. Proposal of the coverage maximization sampling method is not innovative and lacks a detailed explanation in terms of two points. Firstly, the necessity of the sampling strategy requires more explanation: why not just exploit the coverage maximization method as stated in the ER-GNN paper? Moreover, the ER-GNN performs best with the influence maximization (IM) method, which is not mentioned in this paper.
4. In terms of the proposed metrics (Average Accuracy and Average Forgetting), this paper overwhelms other baselines and almost achieves the upper bound. However, the ER-GNN paper adopts different metrics (Performance Mean and Forgeting Mean) which are not inspected in this paper.

**Summary Of The Paper:**

The paper intends to solve the memory explosion problem in the memory replay method for continual graph learning. By presenting Parameter Decoupled Graph Neural Networks (PDGNNs) with the proposed Sufficient Subgraph Embedding Memory (SSEM), the paper shows by empirical studies that the method can reduce memory space by a large margin.

**Summary Of The Review:**

The idea of compressing subgraphs into vectors is straightforward and intuitive, but the method in the paper is composed of off-the-shelf methods and lacks innovation.

---

> ### Author Response · Authors · 2022-11-12
> **Responses to Reviewer zw6P (Part 2)**
>
> **Q3. Proposal of the coverage maximization sampling method is not innovative and lacks a detailed explanation in terms of two points. Firstly, the necessity of the sampling strategy requires more explanation: why not just exploit the coverage maximization method as stated in the ER-GNN paper?**
>
> **A3.** Thanks for this question. Our sampling is developed based on our theoretical findings of pseudo-training effect (Section 3.4, 3.6), and is both theoretically and empirically (Section 4.3) justified to be beneficial for the continual learning performance within our PDGNNs framework.
>
> Although the name is same, our coverage maximization sampling is essentially different from the one in ER-GNN with different designs for different usages. Our method is specially designed for selecting and storing the SSEs generated by our PDGNNs. In contrast, the method stated in ER-GNN paper is designed to select individual nodes. Due to the different targets, our coverage maximization has an entirely different design, as answered in Q2.
>
> In a word, the coverage maximization in ER-GNN is an entirely different method and is not designed for our aim, therefore our design is necessary for our target.
>
> **Q4. Moreover, the ER-GNN performs best with the influence maximization (IM) method, which is not mentioned in this paper.**
>
> **A4.** Thanks for this concern. The code of ER-GNN is not publicly available, therefore we implemented all three different sampling methods in the ER-GNN paper on our own. With our implementations, ER-GNN with CM performs slightly better, so we report the results with CM. The performance of ER-GNN with different sampling methods on different datasets under the class-IL scenario are listed below.
>
> | C.L.T.  | CoraFull | | OGB-Arxiv  | |
> | ---------| ------- |-|-------|-|
> | |AP/\% $\uparrow$ | AF/\% $\uparrow$ | AP/\% $\uparrow$ | AF/\% $\uparrow$ |
> | ER-GNN (MF) | 2.8$\pm$0.1|-94.9$\pm$0.3|12.2$\pm$3.6|-78.8$\pm$4.3|
> | ER-GNN (IM) |2.7$\pm$0.2|-95.1$\pm$0.2|11.8$\pm$2.5|-80.7$\pm$2.7|
> | ER-GNN (CM) |2.9$\pm$0.0|-94.6$\pm$0.1|12.3$\pm$3.1|-79.9$\pm$3.3|
>
> **Q5. In terms of the proposed metrics (Average Accuracy and Average Forgetting), this paper overwhelms other baselines and almost achieves the upper bound. However, the ER-GNN paper adopts different metrics (Performance Mean and Forgetting Mean) which are not inspected in this paper.**
>
> **A5.**  Thanks for this concern. The metrics are identical. In ER-GNN paper, the metrics follow [1], and they explained the metrics as 'To measure the performance in the continual graph learning setup, we take performance mean
> (PM) and forgetting mean (FM) as our evaluation metric. Taking Cora as an example, with
> learning of 3 tasks sequentially, there are 3 accuracy values,
> i.e., one for each task after learning this task; and 3 forgetting
> values, i.e., the performance difference between after learning the particular task and after learning subsequent tasks.'
>
> According to the definitions and the reference on [1], we are sure that our metrics Average Accuracy (AA) and Average Forgetting (AF) are same as the PM and FM in ER-GNN.
>
> Actually these two metrics are widely adopted in all continual learning works [1,2,3,4], although the names are different in different works. For example, they are named as Average Accuracy (ACC) and Backwarde Transfer (BWT) in [1,2], Average Performance (AP) and Average Forgetting (AF) in [3] and Accuracy Mean (AM) and Forgetting Mean (FM) in [4]. In our work, we adopt the names Average Accuracy (AF) and Average Forgetting (AF), which we believe can more accurately convey the meaning of the metrics.
>
> [1] Chaudhry, Arslan, et al. "Riemannian walk for incremental learning: Understanding forgetting and intransigence." Proceedings of the European Conference on Computer Vision (ECCV). 2018.
>
> [2] Lopez-Paz, David, and Marc'Aurelio Ranzato. "Gradient episodic memory for continual learning." Advances in neural information processing systems 30 (2017).
>
> [3] Liu, Huihui, Yiding Yang, and Xinchao Wang. "Overcoming catastrophic forgetting in graph neural networks." Proceedings of the AAAI Conference on Artificial Intelligence. Vol. 35. No. 10. 2021.
>
> [4] Zhang, Xikun, Dongjin Song, and Dacheng Tao. "Hierarchical prototype networks for continual graph representation learning." IEEE Transactions on Pattern Analysis and Machine Intelligence (2022).

---

> ### Author Response · Authors · 2022-11-12
> **Responses to Reviewer zw6P (Part 1)**
>
> We sincerely thank the reviewer for the recognition of our contribution and the constructive comments, which helps us better improve our paper. Our responses to the concerns can be found below.
>
> **Q1. This paper is more like to integrate various existing methods. The formulation of the PDGNN with SSEM is almost identical to that of the SGC (Simplifying Graph Convolutional Networks) method.**
>
> **A1.** Thanks for this concern. First, our contribution is not proposing any specific GNN model, but is to tackle an important problem -how to properly perform continual graph learning with memory replay technique. With our formulated general framework including a restricted GNN framework (PDGNNs) and a memory replay module (SSEM), we successfully resolve the memory explosion problem so that memory replay with topological information is applicable to graph data. The strong experimental results demonstrate that our PDGNNs-SSEM improves the performance significantly compared to existing state-of-the-arts. Besides, we also theoretically reveal a unique phenomenon (pseudo-training effect) in continual graph learning and leverage it to further boost the performance with the novel coverage maximization sampling.
>
> All of these have not been accomplished by existing works and are important issues for continual graph learning.
>
> Moreover, as for the SGC model. Only its neighborhood aggregation is adopted as one candidate instantiation of the function $f_{topo}(\cdot)$ of our PDGNNs, while PDGNNs is a general framework and $f_{topo}(\cdot)$ can also be any arbitrary functions. In experiments, we also adopted three other different instantiations including a non-linear reservoir computing module (Section 3.4). Then the models are entirely different from SGC. Besides, the function $f_{out}(\cdot)$ of PDGNNs can be arbitrary functions and is different from the single fully connected layer of SGC.
>
> Finally, we want to strengthen further that our framework design has significant novelty. The concept of parameter decoupling is specially beneficial for continual graph learning but has never been discussed/discovered in existing works, and we are the first to formulate the framework of parameter decoupled GNNs. Existing decoupled GNNs are not designed for continual learning and focus on other aspects, e.g., decoupling the depth and scope to avoid over-smoothing, etc.
>
> **Q2. The memory replay framework and coverage maximization sampling method almost show no differences from those of the ER-GNN paper**
>
> **A2.** Thanks for this concern. Actually our memory replay mechanism and the coverage maximization sampling are essentially different from the ones in ER-GNN.
>
> First, as for the memory replay part. ER-GNN only stores single nodes for replay, while the topological information is completely ignored. In our work, we encode the computation subgraphs into SSEs, therefore the topological information is preserved. This improvement highly depends on the model structure and can only be achieved in our PDGNNs. In contrast, ER-GNN cannot store topological information because of the memory explosion problem (Section 3.2).
>
> Second, although the names are same (we will consider a better name to differentiate our method from the one of ER-GNN), our coverage maximization is entirely different from the one of ER-GNN. The one in ER-GNN follows the idea to maximally cover the node attribute/embedding space. Specifically, given the attributes/embeddings of a set of nodes, they aim to maximize the mutual distance among the selected subset of nodes to maximally cover the attribute/embedding space. In contrast, our coverage maximization aims to select the SSEs of the nodes with larger computation subgraphs, and does not consider the coverage in attribute/embedding space. Our design is based on our theoretical finding of the pseudo-training effect (Section 3.5,3.6) and is specially beneficial for continual learning.

---

> ### Author Response · Authors · 2022-11-16
> **We are happy to address any remaining/further concern/question from Reviewer zw6P**
>
> We sincerely thank the reviewer for the recognition of our contribution and the constructive comments, which are very helpful to further improve our work.
>
> We have tried our best to address the concerns and questions with both detailed explanations and additional experimental results.
>
> Given the limited time for discussion, we would really appreciate if the reviewer could let us know if the concerns are resolved, and if there is any remaining/additional concern/question. We are more than happy to address any remaining/further concern/question.

---

> ### Author Response · Authors · 2022-12-05
> **Looking forward to post-rebuttal feedback from Reviewer zw6P**
>
> We sincerely thank the reviewer for the recognition of our contribution and the constructive comments, which are very helpful to further improve our work.
>
> We have tried our best to address the concerns and questions with both detailed explanations and additional experimental results.
>
> It has been more 20 days since we submitted our rebuttal, and we would really appreciate if the reviewer could let us know if the concerns are resolved, and if there is any remaining/additional concern/question. We are more than happy to address any remaining/further concern/question.

---

### Official Review · Reviewer_U4Fi · 2022-10-25

**Confidence:** 3
**Correctness:** 4
**Technical Novelty And Significance:** 4
**Empirical Novelty And Significance:** 4
**Recommendation:** 8

**Clarity, Quality, Novelty And Reproducibility:**

The paper is well written. Most sections are well organized. I suggest that you’d better write the contribution of the paper in the form of bullet points, the writing style of the last paragraph of the first section is not easy to read. The contributions of PDGNN with SSEM and coverage-maximization sampling proposed in the paper are significant. The results of the paper can be reproduced by other researchers.

**Details Of Ethics Concerns:**

I don't have any Ethics Concerns.

**Strength And Weaknesses:**

Strengths
The paper is well written.
As mentioned in Section 3.3, the authors noted that NL(v) typically contains O(dL) nodes, replaying n sampled nodes would require storing O(ndL) nodes, where d is the average node degree. In the Reddit dataset, which has an average degree of 492, the buffer size is easily intractable even with a small MPNN. So, directly storing computation subgraphs for memory is not feasible for GNNs. To solve this problem, the proposed method reduces the memory space by storing SSEs instead of the computation subgraphs for optimizing PDGNNs. Specifically, for a model parameterized by θ and the input Gsub v, if optimizing θ with Gsub v or ev is equivalent, then the embedding vector ev is a sufficient subgraph embedding of Gsub v. Recomputing the representation of v every time the trainable parameters are updated requires computing all nodes and edges of Gsub v. To address this issue, the authors propose the Parameter Decoupled Graph Neural Network (PDGNN) framework, which decouples trainable parameters from individual nodes/edges. Its general process is that, first, the topological information of Gsub v is encoded into the embedding vector ev through the unlearnable function ftopo( ). Next, ev is further passed into the trainable function fout(θ) to obtain the output prediction ˆyv, since the trainable parameters act on ev and not directly on any single node/edge, the model parameters θ are optimized using ev or Gsub v are equivalent. Since optimizing the SSE is equivalent to optimizing the computational subgraph of the PDGNN, the memory buffer only needs to store the SSE to reduce the space complexity from O(ndL) to O(n). This is a simple and effective method.
In Section 3.5, the paper mentions that SSE with larger computational graphs covering more nodes may be more efficient. The authors design a coverage-maximization sampling strategy to exploit the benefits of pseudo-training effects. The paper states that computing Rc(SSEM) for all possible SSEs in each iteration is time-consuming, especially on large graphs. Therefore, the authors propose to sample SSEs from a multinomial distribution based on the coverage of each individual SSE. Coverage-maximization sampling strategies appear to be practical.
The authors conducted a number of persuasive experiments. Performances under class-IL are significant.
Related works are properly cited.

Weaknesses
Performance under task-IL is not good enough. It is just a little better than the previous SOTA work.
You'd better write the contribution of the paper in the form of bullet points, the writing style of the last paragraph in the first section is not easy to read.


**Summary Of The Paper:**

The author propose PDGNN with SSEM for continuous graph representation learning to solve technical challenge of applying memory replay techniques to GNNs. By applying SSE, the memory space complexity was reduced, which enables PDGNN to utilize topological information sampled from previous tasks. The author also analyzed the pseudo-training effect of SSE and develop coverage-maximization sampling, which is claimed to be efficient under tight memory budgets.

**Summary Of The Review:**

In conclusion, this is a good paper. I am in favor of accepting this submission.

---

> ### Author Response · Authors · 2022-11-12
> **Responses to Reviewer U4Fi**
>
> We greatly thank the reviewer for the recognition of our contribution and the constructive comments. Our responses are listed below:
>
> **Q1. Performance under task-IL is not good enough. It is just a little better than the previous SOTA work.**
>
> **A1.** Thanks for this concern. Task-IL is much less challenging than class-IL, and the best performing baselines can obtain high performances (around 90%) in task-IL, so that the space for improvement is very limited.
>
> Most importantly, the performance of our PDGNNs-SSEM has actually reached the upper bound of any possible continual learning models, because it is already comparable to Joint (the upper bound on continual learning models).
>
> Therefore, our major experimental contribution lies in the class-IL scenario, and the experiments in task-IL are just for completeness in the settings.
>
> **Q2. You'd better write the contribution of the paper in the form of bullet points, the writing style of the last paragraph in the first section is not easy to read.**
>
> **A2.**
> Thanks for this constructive comment.
> In the updated version, we have carefully rewritten this paragraph and summarized the contributions in the form of bullet points.
>
> This paragraph serves to briefly introduce our motivation and framework design. First, since the key challenge is the unbounded size of the computation subgraph, we desire to encode each computation subgraph into a fixed size vector, i.e. the sufficient subgraph embedding (SSE). Then, we explain the structural requirements of deriving SSEs, and accordingly formulate the PDGNNs framework. After introducing the framework, we introduced the theoretical findings under this framework, and the accordingly developed sampling strategy (coverage maximization sampling) to better populate the memory buffer. Finally, we introduced the adopted continual learning scenarios (class-IL and task-IL).

---

### Official Review · Reviewer_AL85 · 2022-10-27

**Confidence:** 4
**Correctness:** 3
**Technical Novelty And Significance:** 2
**Empirical Novelty And Significance:** 2
**Recommendation:** 5

**Clarity, Quality, Novelty And Reproducibility:**

The paper is clearly written and easy to read. There lacks some technical depth in the proposed model as I feel it is more like applying decoupled GNNs to the existing continual learning pipeline. The Theoretical results also lack some significance. Overall, the novelty seems to be limited due to its close connection with known decoupled and continual learning models.

**Strength And Weaknesses:**

## Strengths

+ The paper is clearly written and easy to read.
+ The authors sufficiently discuss the related works in various areas, including continual learning and decoupled models.
+ The proposed method achieves significant improvements in empirical evaluation.


## Weaknesses

- The novelty is limited. The overall idea is simple. I view it as applying decoupled GNN models to the existing continual learning pipeline. The decoupled GNNs are based on known models in the literature.
- The theoretical results are not too significant. The usefulness of pseudo-training highly depends on the extent of homophily as well as the graph structure. If a neighbor node has a different label from the target node, then the pseudo-training effect is actually not desired. In addition, if the neighborhood of a neighbor node is different from that of a target node, then the pseudo-training will also deviate significantly from the “actual” training.
- How do you interpret the re-scaling factor in Theorem 1. For example, why such specific form of scaling helps improve learning quality? Why it makes sense to train the "real" target node without re-scaling yet pseudo-train the neighbor nodes with re-scaling?
- Although empirically working well, the proposed coverage maximization sampling is based on a straightforward heuristic.
- Baselines in experiments seem to be a bit out-dated. Most of them are 2017 and 2018 models. In addition, from Table 2, it seems that coverage maximization does not significantly outperform other sampling algorithms, even though the coverage ratio of other sampling algorithms is significantly lower. The leads to a question on how important is the “coverage maximization” criteria.


**Summary Of The Paper:**

This paper presents a memory replay-based continual learning model based on simplified GNN models. When new nodes are added to the graph, the continual learning GNN keeps optimizing on both the new nodes and the old nodes in the reply buffer to avoid forgetting the historical information. The decoupling of message propagation from feature transformation enables the replay buffer to only store the target node embeddings without any neighborhood information, thus addressing the challenge of neighborhood explosion. Essentially, the proposed method is to apply decoupled GNN models to existing continual graph learning frameworks. Experiments show significant accuracy gains compared with baseline continual graph learning methods.


**Summary Of The Review:**

Overall, although this is a well-written paper, I think it is still below the bar of acceptance due to its limited novelty and technical depth.

---

> ### Author Response · Authors · 2022-11-12
> **Responses to Reviewer AL85 (Part 3)**
>
> **Q3. How do you interpret the re-scaling factor in Theorem 1. For example, why such specific form of scaling helps improve learning quality? Why it makes sense to train the "real" target node without re-scaling yet pseudo-train the neighbor nodes with re-scaling?**
>
> **A3.** Thanks for the question. The re-scaling factor is not something we designed to improve the learning quality. Instead, it is simply an algebraic fact. Specifically, replaying an SSE to the model (retraining the model with the SSE) is equivalent to a pseudo-training on the neighbors. But this pseudo-training effect does not equally influence different neighbors. It has different influence strengths on different neighbors. As shown in Proof of Theorem 1.2 (Appendix B.2, page 16), with function $f_{out}(\cdot)$ being a linear layer, the strength is determined by multiple factors and are summarized as the re-scaling factor after algebraic calculation.
>
> **Q4. Although empirically working well, the proposed coverage maximization sampling is based on a straightforward heuristic.**
>
> **A4.**
> Thanks for the recognition of the empirical effectiveness of the proposed coverage maximization sampling.
>
> Our coverage maximization has a solid motivation based on theoretical findings. As explained in Section 3.6, due to the existence of pseudo-training effect (Section 3.5), selecting the SSEs of the nodes with more neighbors may benefit the continual learning performance on homophilic graphs. Accordingly, we develop the coverage maximization sampling aiming to select the nodes with more neighbors.
>
> The analysis is justified by multiple experiments with different memory budget configurations. Results show that coverage maximization sampling indeed improves the performance especially when the memory budget is tight, which is practical in real-world scenarios (Table 2. Section 4.3).
>
>
> **Q5. Baselines in experiments seem to be a bit out-dated. Most of them are 2017 and 2018 models.**
>
> **A5.**
> Thanks for this concern. This is a misconception. Actually, 2 out of 6 baselines are 2021 models, and the other 4 baselines are representative methods which are also adopted by other continual graph learning works [1,2].
>
> Most importantly, our baselines include joint training (Joint), which is the performance upper bound. Since no other baselines would outperform joint training, the baselines we adopted are enough to demonstrate the performance superiority of our PDGNNs-SSEM.
>
> [1] Liu, Huihui, Yiding Yang, and Xinchao Wang. "Overcoming catastrophic forgetting in graph neural networks." Proceedings of the AAAI Conference on Artificial Intelligence. Vol. 35. No. 10. 2021.
>
> [2] Zhang, Xikun, Dongjin Song, and Dacheng Tao. "Hierarchical prototype networks for continual graph representation learning." IEEE Transactions on Pattern Analysis and Machine Intelligence (2022).
>
> **Q6. In addition, from Table 2, it seems that coverage maximization does not significantly outperform other sampling algorithms, even though the coverage ratio of other sampling algorithms is significantly lower. The leads to a question on how important is the “coverage maximization” criteria.**
>
> **A6.** Thanks for this question. First, Coverage maximization sampling is important when memory budget is tight, as shown in Table 2. In Table 2, we show a wide range of choices including not only tight budgets but also large budgets for a comprehensive comparison. But in real-world applications, tight budgets is much more practical, and storing more than 1.0\% data may be infeasible especially on large graphs with millions of nodes. Therefore, our coverage maximization sampling is of great importance for practical applications.
>
> Second, please note that the determinant mechanism of our PDGNNs-SSEM to obtain superior performance is the design that successfully stores and replay topological information, while the sampling strategy is the icing on the cake. Therefore, the fact that all different sampling methods can obtain satisfying results actually reveal the robustness of PDGNNs-SSEM, and is an advantage of PDGNNs-SSEM.

---

> ### Author Response · Authors · 2022-11-12
> **Responses to Reviewer AL85 (Part 2)**
>
> **Q2. The theoretical results are not too significant. The usefulness of pseudo-training highly depends on the extent of homophily as well as the graph structure. If a neighbor node has a different label from the target node, then the pseudo-training effect is actually not desired. In addition, if the neighborhood of a neighbor node is different from that of a target node, then the pseudo-training will also deviate significantly from the “actual” training.**
>
> **A2.** Thanks for this concern. First, if the situation mentioned by the reviewer is prevalent in the graph, then the graph is a heterophilic graph. Homophilic and heterophilic graphs require essentially different GNN designs  [1,2,3], and it is improper to apply homophilic GNNs to heterophilic graphs. In this work, all of our data are homophilic graphs and our model instantiations also target homophilic graphs. In general, the situation mentioned by the reviewer should not be prevalent in homophilic graphs, and most neighbors will benefit from pseudo-training, which is justified by our strong experimental results on different large homophilic graph datasets.
>
> When heterophilic graphs are given, the function $f_{topo}(\cdot)$ of our PDGNNs should be modified accordingly (just like homophilic GNNs have to be specially adapted for learning on heterophilic graphs), and pseudo-training effect can still be beneficial. In the following, we will first introduce how to construct $f_{topo}(\cdot)$ for heterophilic graphs (the next 3 paragraphs), and then explain why pseudo-training still benefits (last paragraph of this answer).
>
> The key difference of heterophilic graph is that the nodes of the same class are not likely to be connected, and GNNs should be designed to separately process the proximal neighbors with similar information and distal neighbors with irrelevant information, and only aggregate information from the proximal neighbors [1,2,3].
>
> Accordingly, the first strategy to construct $f_{topo}(\cdot)$ following the MixHop [2] and let $f_{topo}(\cdot)$ encode neighbors from different hops separately. Specifically, the computation subgraph, which is the input to $f_{topo}(\cdot)$, will be divided into different hops. For neighbors in each hop, $f_{topo}(\cdot)$ generates an embedding with a certain strategy. Finally, the embeddings of different hops are concatenated (summation should be avoided to ensure different hops are processed separately) as the final SSE (Equation 4 in the paper). If some hops (e.g. 1-hop) always contain irrelevant information, their contribution can become arbitrarily small, which is controlled by the trainable weights in function $f_{out}(\cdot)$ of PDGNNs.
>
> The second strategy follows H2GCN [3] to only aggregate higher-order neighbors, because H2GCN [3] theoretically justifies that two-hop neighbors tend to be proximal to the center node, if the the one-hop neighbors have labels that are conditionally independent of the center node's label. In other words, for designing $f_{topo}(\cdot)$, the one-hop neighbors can simply be ignored during neighborhood aggregation.
>
> In this way, by constructing $f_{topo}(\cdot)$ to be suitable for heterophilic graphs, the information aggregation is conducted on the proximal neighbors. Accordingly, given a target node $v$, if its SSE is stored and replayed, the pseudo-training will only influence the proximal neighbors of $v$ (denoted as $N_p(v)$ for convenience). Then the reviewer's concern 'a neighbor node has a different label from the target node' is resolved. As for the second concern 'if the neighborhood of a neighbor node is different from that of a target node'. Since nodes in $N_p(v)$ also aggregate proximal neighbors during actual training, no matter the neighborhood of nodes in $N_p(v)$ is proximal to $v$ or not, the neighborhood information received by nodes in $N_p(v)$ are similar to the neighborhood information received by $v$. In other words, the node embedding and labels for nodes in  $N_p(v)$ during pseudo-training are indeed similar to their embedding in actual training due to the heterophilic GNN design.
>
> Incorporating heterophilic graphs into continual graph learning is promising and interesting. In our future works, we will accordingly construct continual learning tasks on heterophilic graphs and implement suitable models as introduced above.
>
> [1] Zheng, Xin, et al. "Graph neural networks for graphs with heterophily: A survey." arXiv preprint arXiv:2202.07082 (2022).
>
> [2] Abu-El-Haija, Sami, et al. "Mixhop: Higher-order graph convolutional architectures via sparsified neighborhood mixing." international conference on machine learning. PMLR, 2019.
>
> [3] Zhu, Jiong, et al. "Beyond homophily in graph neural networks: Current limitations and effective designs." Advances in Neural Information Processing Systems 33 (2020): 7793-7804.

---

> ### Author Response · Authors · 2022-11-12
> **Responses to Reviewer AL85 (Part 1)**
>
> We sincerely thank the reviewer for recognizing our contribution and the constructive comments. The answers to the concerns are listed below.
>
> **Q1. The novelty is limited. The overall idea is simple. I view it as applying decoupled GNN models to the existing continual learning pipeline. The decoupled GNNs are based on known models in the literature.**
>
> **A1.** Thanks for this concern. First, please note that our contribution is not proposing a specific novel GNN, but is to tackle an important problem - resolving continual graph learning with memory replay. With our formulated general framework including a restricted GNN framework (PDGNNs) and a memory replay module (SSEM), we successfully resolve the memory explosion problem so that memory replay with explicit topological information is enabled on graph data. The strong experimental results demonstrate that our PDGNNs-SSEM improve the performance significantly compared to existing state-of-the-arts. Besides, we also theoretically reveal a unique phenomenon (pseudo-training effect) in continual graph learning, and leverage it to further boost the performance with the novel coverage maximization sampling.
>
> All of these have not been accomplished by existing works and are important contributions for continual graph learning.
>
> Moreover, our model design also has significant novelty. The concept of parameter decoupling is especially beneficial for continual graph learning but has never been discussed in existing works, and we are the first to formulate the framework of parameter decoupled GNNs. Existing decoupled GNNs are not designed for continual learning and focus on different aspects, e.g., decoupling the depth and scope to avoid over-smoothing, etc.

---

> ### Author Response · Authors · 2022-11-16
> **We are happy to address any remaining/further concern/question from Reviewer AL85**
>
> We sincerely thank the reviewer for the recognition of our contribution and the constructive comments, which are very helpful to further improve our work. We have tried our best to address the concerns and questions.
>
> Given the limited time for discussion, we would really appreciate if the reviewer could let us know if the concerns are resolved, and if there is any remaining/additional concern/question. We are more than happy to address any remaining/further concern/question.

---

> ### Author Response · Authors · 2022-12-05
> **Looking forward to post-rebuttal feedback from Reviewer AL85**
>
> We sincerely thank the reviewer for the recognition of our contribution and the constructive comments, which are very helpful to further improve our work. We have tried our best to address the concerns and questions.
>
> It has been more 20 days since we submitted our rebuttal, and we would really appreciate if the reviewer could let us know if the concerns are resolved, and if there is any remaining/additional concern/question. We are more than happy to address any remaining/further concern/question.

---

### Official Review · Reviewer_g88c · 2022-10-31

**Confidence:** 4
**Correctness:** 4
**Technical Novelty And Significance:** 2
**Empirical Novelty And Significance:** 2
**Recommendation:** 3

**Clarity, Quality, Novelty And Reproducibility:**

The paper is clear and well written.

Reproducibility of the experiments is difficult due to the lack of all the implementation details (a link to the code for example).




**Details Of Ethics Concerns:**

I didn't find any ethical issues.

**Strength And Weaknesses:**

*Strength*

The paper aims to suggest solutions for an important real-world problem.

The suggested method outperforms previously suggested techniques for continual learning.

*Weaknesses*

The contributions for the paper, as presented by the authors, are divided into three:

1. PDGNNs - SSEs
2. Pseudo-training effect discovery
3. coverage maximization sampling strategy

I will refer to each of them separately.

1. The idea of decoupling the weights from the topology using two steps, in which nodes are represented as vectors, is trivial. I believe the challenge is how to encode the nodes to minimize the effect of the forgetting problem. The authors chose to encode the entire subgraph required by an MPGNN with L layers to represent the node as a vector. Their suggested technique is strongly based on previous works with minor additions. Justifications for the additions are also missing (for example, why the summation operation is being used). Moreover, different works encode subgraphs in an unsupervised manner [1,2], so a comparison is required here, in my opinion.

2. The pseudo-training effect is an expected phenomenon when dealing with homophilic graphs.
Question: Will the pseudo-training effect damage the performance of PDGNN - SSE on heterophilic graphs?

3. The sampling method is relatively naïve, and it is not compared to any complicated sampling method presented in the literature [3,4].
Question: What changes (if any) are needed to be done to fit the sampling method to heterophilic graphs?



Question: In table 4, why PDGNNs sometimes outperform Joint?


[1] - Sub2Vec: Feature Learning for Subgraphs, Adhikari et al.
[2] - subgraph2vec: Learning Distributed Representations of Rooted Sub-graphs from Large Graphs, Narayanan et al.
[3] - Metropolis Algorithms for Representative Subgraph Sampling, H¨ubler et al.
[4] - Diversified Top-k Subgraph Querying in a Large Graph, Yang et al.

**Summary Of The Paper:**

The paper suggests a method for buffer construction in memory replay for node classification.
The suggested technique reduce the space complexity of the trivial algorithm from O(nd^L) to O(n).
The authors also discover and define the pseudo-training effect which is unique to continual learning on non-euclidean data. Finally, the authors present a coverage maximization sampling strategy to improve their suggested continual learning technique further.

**Summary Of The Review:**

From the reasons written above, I think that the novelty of the paper is relatively weak and not meets the threshold of being published in a top-tier venue.
Hence my pre-rebuttal score is (reject: 3)

I expect the authors to respond to my concerns raised in the weaknesses section for my re-evaluation of this manuscript.

---

> ### Author Response · Authors · 2022-11-12
> **Responses to Reviewer g88c (Part 5)**
>
> **Q7. What changes (if any) are needed to be done to fit the sampling method to heterophilic graphs?**
>
> **A7.** Thanks for this question. To fit the heterophilic graphs, as answered in Q5, modifications should be done to function $f_{topo}(\cdot)$ instead of the sampling method, which only serves populate the memory buffer and is does not determine how the model process input graph data.
> The rationale here is same as that the classic GNNs (MPNNs) for homophilic graph learning perform badly on heterophilic graphs. And the heterophilic GNNs require essentially different model structures [1,2,3]. Details have been answered in Q5.
>
> Details of how to modify $f_{topo}(\cdot)$ for heterophilic graphs are also answered in Q5. Incorporating heterophilic graphs is a very promising direction and will be our future works.
>
> [1] Zheng, Xin, et al. "Graph neural networks for graphs with heterophily: A survey." arXiv preprint arXiv:2202.07082 (2022).
>
> [2] Abu-El-Haija, Sami, et al. "Mixhop: Higher-order graph convolutional architectures via sparsified neighborhood mixing." international conference on machine learning. PMLR, 2019.
>
> [3] Zhu, Jiong, et al. "Beyond homophily in graph neural networks: Current limitations and effective designs." Advances in Neural Information Processing Systems 33 (2020): 7793-7804.
>
> **Q8. In table 4, why PDGNNs sometimes outperform Joint?**
>
> **A8.** Thanks for this question. First, the reason Joint (joint training) is regarded as the upper bound is because it learns all tasks simultaneously without the forgetting problem. However, the forgetting problem is not the only factor determining the performance. Therefore, when the forgetting problem is well addressed (like our proposed PDGNNs-SSEM), the influence of other factors would emerge. Specifically,
>
> 1. PDGNNs-SSEM learn tasks sequentially, the learning starts from few classes and gradually learn more new classes in the new tasks. In contrast, Joint learns all classes (tasks) simultaneously. These two learning manners result in different optimization difficulties. For datasets whose task sequence happen to contain easy-to-hard sub-sequences, learning the tasks sequentially is easier than learning jointly. This phenomenon is also studied in existing works, e.g.  [1]. Therefore, when the forgetting problem is well addressed and is not the dominating factor, PDGNNs may outperform joint training when sequential learning is more suitable for the given dataset.
>
> 2. When learning a task in the task sequence, information of the previous tasks is provided to PDGNNs through the stored SSEs, which are representative data selected from the original data. While for Joint, all of the original data are used, which may contain noisy data that are detrimental to the performance in sometime. In other words, noise may be reduced by our proposed coverage max sampling, and replaying the selected SSEs is better than training with the complete original data.
>
> [1] Bhat, S. Divakar, et al. "CILEA-NET: Curriculum-Based Incremental Learning Framework for Remote Sensing Image Classification." IEEE Journal of Selected Topics in Applied Earth Observations and Remote Sensing 14 (2021): 5879-5890.

---

> ### Author Response · Authors · 2022-11-12
> **Responses to Reviewer g88c (Part 4)**
>
> **Q6. The sampling method is relatively naive, and it is not compared to any complicated sampling method presented in the literature [1,2]**
>
> **A6.**
> Thanks for this concern. First, the coverage maximization sampling is inspired by our theoretical finding of pseudo-training effect (Section 3.5, 3.6). It is specially designed to benefit the continual learning performance studied in our work. Moreover, multiple experiments have been conducted to demonstrate that coverage maximization sampling indeed brings improvements especially when the memory budget is tight (Table 2, Section 4.3), which is practical in real-world applications.
>
> In contrast, the complicated sampling methods in standard learning are not specialized for continual learning and have different objectives (e.g. approximating the original graph in terms of certain statistics [1], etc.). Therefore, these methods may not be suitable for continual learning. Moreover, being too complicated brings down computational efficiency, which is an important aspect of a continual learning model that often encounters new tasks.
>
> The two papers [1,2] recommended by the reviewer are very interesting works and we have added discussion in our paper (Section 4.3). Details of their applicability to our task are listed below.
>
> The first work [1] proposes the Metropolis Algorithms for subgraph sampling and is applicable to our work. We implemented this method (with the degree distribution distance function of [1]) and tested it on the OGB-Arxiv dataset under the class-IL scenario with different memory budgets, following the setting in Table 2 of our paper. The results are shown below.
>
> | Ratio of dataset / \% | 0.02 | 0.1 | 1.0  | 5.0 | 40.0 |
> | ---------| ------- |-|-------|-|-|
> | Uniform sampling |12.0$\pm$1.1|24.1$\pm$1.7|42.2$\pm$0.3|50.4$\pm$0.4|53.3$\pm$0.4 |
> | Mean of feature | 12.6$\pm$0.1|25.3$\pm$0.3|42.8$\pm$0.3|50.4$\pm$0.7|53.3$\pm$0.2 |
> | Metropolis Algorithms [1] |11.8$\pm$0.4|23.2$\pm$1.3|41.4$\pm$1.1|48.3$\pm$0.4|52.3$\pm$0.1|
> | Coverage Maximization |14.9$\pm$0.8|26.8$\pm$1.8|43.7$\pm$0.5|50.5$\pm$0.4|53.4$\pm$0.1|
>
> According to the table above, Metropolis Algorithms does not exhibit superiority over the other sampling methods including our coverage maximization sampling. This result is reasonable since Metropolis Algorithms is not designed for continual learning and does not consider how to better populate the memory buffer to better avoid forgetting. In contrast, based on theoretical findings, our coverage maximization sampling is specially designed to select the nodes that would better avoid the forgetting problem. Besides, Metropolis Algorithms require complicated computation of the distance between the sampled graph and the original graph, therefore are less efficient than the algorithms adopted in our work.
>
> Moreover, please note that the key mechanism in our PDGNNs to obtain superior performance is the design to successfully store topological information in the memory buffer (one of our major contributions), while the sampling strategy (how to populate the buffer) is the icing on the cake. Therefore, experimental results (Table 2 of Section 4.3, and Table 1 in this answer) have demonstrated that whatever sampling method is used, our framework can perform very well. Coverage maximization sampling, which is specially for continual learning, will make the performance even better, especially with a tight memory budget (a very practical scenario).
>
> Finally, [2] is designed to find the top-k isomorphic subgraphs with maximal diversity given an query subgraph. It is not a sampling method and is inapplicable to our task. What we desire is to sample a set of SSEs (nodes). There is not any query in our task, and the isomorphic subgraph matching is not relevant either.
>
> [1] - Metropolis Algorithms for Representative Subgraph Sampling, H¨ubler et al.
>
> [2] - Diversified Top-k Subgraph Querying in a Large Graph, Yang et al.

---

> ### Author Response · Authors · 2022-11-12
> **Responses to Reviewer g88c (Part 3)**
>
> **Q5. The pseudo-training effect is an expected phenomenon when dealing with homophilic graphs. Question: Will the pseudo-training effect damage the performance of PDGNN - SSE on heterophilic graphs?**
>
> **A5.** Thanks for this question. Briefly speaking, as long as $f_{topo}(\cdot)$ is properly constructed, pseudo-training will not damage the performance.
>
> Heterophilic graph learning is largely different from homophilic graph learning, and requires different GNN designs [1]. Therefore, for learning on heterophilic graphs, the function $f_{topo}(\cdot)$ of PDGNNs should also be instantiated to be suitable for heterophilic graphs. The rationale here is same as that the classic GNNs (MPNNs) for homophilic graphs perform badly on heterophilic graphs. And the heterophilic GNNs require essentially different model structures [1,2,3].
>
> In this work, we focus on homophilic graphs, and all instantiations of $f_{topo}(\cdot)$ are designed for homophilic graphs.
> However, instantiating $f_{topo}(\cdot)$ to be suitable for heterophilic graphs is straightforward.
>
> In the following, we will first explain how to configure $f_{topo}(\cdot)$ for heterophilic graphs, and then explain why it ensures that pseudo-training will not damage the performance.
>
> The key difference of heterophilic graph learning is that the nodes belonging to the same classes are not likely to be connected, and GNNs should be designed to separately process the proximal neighbors with similar information and distal neighbors with irrelevant information, or only aggregate information from the proximal neighbors [1,2,3].
>
> Accordingly, the first strategy to construct $f_{topo}(\cdot)$ is to follow the MixHop [2] and let $f_{topo}(\cdot)$ encodes neighbors from different hops separately. Specifically, a given computation subgraph (the input to $f_{topo}(\cdot)$) will be divided into different hops. For neighbors in each hop, $f_{topo}(\cdot)$ generates an embedding with a certain strategy. Finally, the embeddings of different hops are concatenated (summation should not be used to ensure different hops are separately processed) as the final SSE (Equation 4 in the paper).
>
> The second strategy follows H2GCN [3] to only aggregate higher-order neighbors, because H2GCN [3] theoretically justifies that two-hop neighbors tend to be proximal to the center node, if the the one-hop neighbors have labels that are conditionally independent of the center node's label. In other words, for designing $f_{topo}(\cdot)$, the one-hop neighbors can be simply ignored when doing neighborhood aggregation.
>
> The reviewer's concern that 'pseudo-training may damage the performance on heterophilic graphs' should be because the pseudo-training may be conducted on the distal nodes containing irrelevant information. But via constructing $f_{topo}(\cdot)$ to be suitable for heterophilic graphs, the neighborhood aggregation is still conducted on the proximal nodes, and so is the pseudo-training. In this way, the pseudo-training will not damage but still benefit the performance.
>
> Incorporating heterophilic graphs into continual graph learning is promising and interesting. In our future works, we will construct continual learning tasks on heterophilic graphs and implement suitable models as introduced above.
>
> [1] Zheng, Xin, et al. "Graph neural networks for graphs with heterophily: A survey." arXiv preprint arXiv:2202.07082 (2022).
>
> [2] Abu-El-Haija, Sami, et al. "Mixhop: Higher-order graph convolutional architectures via sparsified neighborhood mixing." international conference on machine learning. PMLR, 2019.
>
> [3] Zhu, Jiong, et al. "Beyond homophily in graph neural networks: Current limitations and effective designs." Advances in Neural Information Processing Systems 33 (2020): 7793-7804.

---

> ### Author Response · Authors · 2022-11-12
> **Responses to Reviewer g88c (Part 2)**
>
> **Q3. Justifications for the additions are also missing (for example, why the summation operation is being used).**
>
> **A3.** Thanks for this concern. First, the summation operation is just one potential realization of the function $f_{topo}(\cdot)$. PDGNNs is a general framework, and the function $f_{topo}(\cdot)$ can be arbitrary functions. Accordingly, we already included experimental results of constructing $f_{topo}(\cdot)$ with non-linear functions (reservoir computing module) instead of summation (Section 3.4, Appendix C.3).
>
> Second, the justification for using the summation is explained in Section 3.4, and includes the following aspects. Experimentally, the linear summation formulation performs comparably to the complex non-linear formulations but is more efficient. Theoretically, rigorous analysis can be developed (Section 3.5) and is used to further improve the performance (Section 3.6).
>
>
> **Q4. Moreover, different works encode subgraphs in an unsupervised manner [1,2], so a comparison is required here, in my opinion.**
>
> **A4.**
> We sincerely thank the reviewer for raising this point. By clarifying this point, the key reason that naive memory replay cannot be applied to GNNs, as well as the rationale of our parameter decoupling design, will be easy to understand.  We have also added a discussion on [1,2] in our paper (Section 3.2) for further clarification.
>
> First of all, constructing the first step of PDGNNs (function $f_{topo}(\cdot)$) as an unsupervised learning module would introduce trainable parameters to $f_{topo}(\cdot)$. Since the data of each task cannot be known in advance, unsupervised learning have to be conducted whenever a new task comes. After learning each new task, the parameters of $f_{topo}(\cdot)$ would change, and its generated embeddings (SSEs) of nodes in previous tasks also change. Originally, we would store these SSEs for memory replay, but now these SSEs become obsolete every time a new task is learned and cannot be used to retrain the model.
>
> In other words, adopting an unsupervised learning module is incompatible with our memory replay mechanism.
>
> To experimentally demonstrate this point, we implemented $f_{topo}(\cdot)$ as the unsupervised learning model Sub2Vec [1] recommended by the reviewer, and conduct experiments on OGB-Arxiv and CoraFull datasets under the class-IL scenario. Specifically, Sub2Vec encodes a given subgraph into a vector. In our implementation, we will feed the computational subgraph around each node into Sub2Vec to get an embedding for each node. As shown in the table below, adopting Sub2Vec results in better performance than Fine-tune based model without continual learning technique, but still exhibit severe forgetting problem because now the memory replay part is invalid.
>
> | C.L.T.  | CoraFull | | OGB-Arxiv  | |
> | ---------| ------- |-|-------|-|
> | |AP/\% $\uparrow$ | AF/\% $\uparrow$ | AP/\% $\uparrow$ | AF/\% $\uparrow$ |
> | Fine-tune| 3.5$\pm$0.2|-95.2$\pm$0.3 | 4.9$\pm$0.0|-89.7$\pm$0.4 |
> | PDGNNs-SSEM (Sub2Vec) |12.5$\pm$3.2|-78.2$\pm$1.4 | 10.0$\pm$3.1|-80.7$\pm$4.1 |
> | Original PDGNNs-SSEM   |81.9$\pm$0.1|-3.9$\pm$0.1 | 53.2$\pm$0.2|-14.7$\pm$0.2|
>
> [1] - Sub2Vec: Feature Learning for Subgraphs, Adhikari et al.
>
> [2] - subgraph2vec: Learning Distributed Representations of Rooted Sub-graphs from Large Graphs, Narayanan et al.

---

> ### Author Response · Authors · 2022-11-12
> **Responses to Reviewer g88c (Part 1)**
>
> We sincerely appreciate the reviewer for recognizing the importance of the problem studied in our work and our technical as well as empirical contributions. Our responses to the concerns are listed below.
>
> **Q1. The idea of decoupling the weights from the topology using two steps, in which nodes are represented as vectors, is trivial. I believe the challenge is how to encode the nodes to minimize the effect of the forgetting problem.**
>
> **A1.** Thanks for this concern. We totally agree that the challenge is to minimize the effect of forgetting. That is exactly why we decouple the weights.
>
> Specifically, memory replay is currently the most effective technique for mitigating the forgetting problem in traditional continual learning (on independent data without explicit topological connections), especially in the challenging class-IL scenario [1,2,3,4,5]. However, in continual graph learning, due to the memory explosion problem (Section 3.2), valuable explicit topological information cannot be properly stored and replayed. Because of this, existing models cannot perform well in the challenging class-IL scenario, as shown in Table 3.
>
> Accordingly, our decoupling architecture is specially designed to tackle this challenge and enable effective memory replay by incorporating explicit topological information, which has never been achieved in existing works.
>
> Based on our proposed decoupling strategy, memory replay with explicit topological information is achieved for the first time, and the forgetting problem is resolved reasonably well, as demonstrated by the strong experimental results (Table 3).
>
> Therefore, we respectively disagree that the weight decoupling step is trivial. This is because it serves as a critical step to facilitate memory replay on graphs and mitigate the forgetting problem accordingly.
>
> [1] Van de Ven, Gido M., and Andreas S. Tolias. "Three scenarios for continual learning." arXiv preprint arXiv:1904.07734 (2019).
>
> [2] Rebuffi, Sylvestre-Alvise, et al. "icarl: Incremental classifier and representation learning." Proceedings of the IEEE conference on Computer Vision and Pattern Recognition. 2017.
>
> [3] Prabhu, Ameya, Philip HS Torr, and Puneet K. Dokania. "Gdumb: A simple approach that questions our progress in continual learning." European conference on computer vision. Springer, Cham, 2020.
>
> [4] Bang, Jihwan, et al. "Rainbow memory: Continual learning with a memory of diverse samples." Proceedings of the IEEE/CVF Conference on Computer Vision and Pattern Recognition. 2021.
>
> [5] Kim, Chris Dongjoo, et al. "Continual learning on noisy data streams via self-purified replay." Proceedings of the IEEE/CVF international conference on computer vision. 2021.
>
> **Q2. The authors chose to encode the entire subgraph required by an MPGNN with L layers to represent the node as a vector. Their suggested technique is strongly based on previous works with minor additions.**
>
> **A2.** Thanks for this concern. First, although the instantiations of the function $f_{topo}(\cdot)$ in PDGNNs can be based on previous works, it is not claimed as our contribution. Instead, our contribution and novelty lie in successfully solving an important research problem - resolving continual graph learning with memory replay technique.
>
> Specifically, we proposed the framework to tackle the memory explosion problem and effectively apply memory replay to continual graph learning. The strong experimental results demonstrate our success, especially in the challenging class-IL scenario (Table 3). Based on this framework, we also theoretically analyzed the pseudo-training effect that is unique in continual graph learning and developed a novel coverage maximization sampling to leverage this effect. These have never been achieved in existing works and therefore are significant contributions as recognized by reviewer U4Fi.
>
> Moreover, as for the dependency on previous works. The general concept of parameter decoupling was never discussed by existing works. We are the first to formulate a general framework for parameter decoupled GNNs (existing works [1,2,3] can be treated as instantiations of the framework), which is especially suitable for continual graph learning. Existing decoupled GNNs focus on other aspects, e.g., decoupling the depth and scope to avoid over-smoothing, etc.
>
> [1] Wu, Felix, et al. "Simplifying graph convolutional networks." International conference on machine learning. PMLR, 2019.
>
> [2] Zhu, Hao, and Piotr Koniusz. "Simple spectral graph convolution." International Conference on Learning Representations. 2020.
>
> [3] Gallicchio, Claudio, and Alessio Micheli. "Fast and deep graph neural networks." Proceedings of the AAAI conference on artificial intelligence. Vol. 34. No. 04. 2020.

---

> ### Author Response · Authors · 2022-11-16
> **We are happy to address any remaining/further concern/question from Reviewer g88c**
>
> We sincerely thank the reviewer for the recognition of our contribution and the valuable comments, which are very constructive to further improve our work.
>
> We have tried our best to address the concerns and questions with detailed explanations, additional experiments, and additional discussions added to the paper.
>
> Given the limited time for discussion, we would really appreciate if the reviewer could let us know if the concerns are resolved, and if there is any remaining/additional concern/question. We are more than happy to address any remaining/further concern/question.

---

> ### Author Response · Authors · 2022-12-05
> **Looking forward to post-rebuttal feedback from Reviewer g88c**
>
> We sincerely thank the reviewer for the recognition of our contribution and the valuable comments, which are very constructive to further improve our work.
>
> We have tried our best to address the concerns and questions with detailed explanations, additional experiments, and additional discussions added to the paper.
>
> It has been more 20 days since we submitted our rebuttal, and we would really appreciate if the reviewer could let us know if the concerns are resolved, and if there is any remaining/additional concern/question. We are more than happy to address any remaining/further concern/question.

---

### Author Response · Authors · 2022-11-18
**A friendly reminder that the discussion is ending in one day**

Again, we sincerely thank all reviewers for the recognition of our contribution and the valuable comments, which are very constructive to further improve our work. To further facilitate the reviewing process, we briefly summarize the reviews and our responses below. And we would really appreciate if the reviewers could let us know if the concerns are resolved.

1. Regarding how to adapt our PDGNNs for heterophilic graphs and whether the pseudo-training effect can still be beneficial (g88c,AL85), we have provided very specific strategies for model adaptation, as well as detailed explanations on how is pseudo-training still beneficial on heterophilic graphs.

2. Regarding the connections and similarity between our PDGNNs and some existing works (g88c,AL85,zw6P), we have explained that our main contribution is tackling the challenging memory explosion problem and successfully applying memory replay to graph data, instead of proposing any specific model. Moreover, we also explained that our PDGNNs are essentially different from existing works in terms of both the target and model design.

3. Regarding the suggestion to compare with some unsupervised models and complicated sampling methods (g88c), we have provided additional experimental results as well as detailed analysis on the applicability of different methods.

4. Regarding the questions on our coverage maximization strategy (AL85, zw6P), we provided detailed explanations on its motivation, differentiation, and suitable application scenarios.

5. Regarding all the other miscellaneous concerns, we have also provided very detailed explanations to clarify each point and made revision in the paper whenever necessary.

---

### Decision · Program_Chairs · 2023-01-20

**Decision:**

Reject

**Justification For Why Not Higher Score:**

The main issues (in order) were the limited technical novelty (compared to decoupled GNNs, which have well established benefits in computational efficiency), the suitability of the method's design for heterophilic graphs, and evidence for benefits of the coverage maximization strategy.

**Justification For Why Not Lower Score:**

N/A

**Metareview: Summary, Strengths And Weaknesses:**

The paper proposes an approach for continual learning in the graph learning setting, based on a memory replay approach. To avoid the neighborhood explosion issue where too many node embeddings must be stored, they develop a decoupled approach (PDGNN) to enable efficient memory replay. They then propose a pseudo-training effect which strengthens the prediction of nodes in the same subgraph, motivating a coverage maximization sampling strategy to enhance performance.

In general, reviewers find the paper to be clearly written, and well-motivated by the memory explosion issue. However, there are a number of main issues raised by reviewers:

- Some reviewers felt that the technical novelty is limited, particularly as the method can be seen as an application of decoupled GNNs to the continual learning setting, considering that decoupled GNNs have well established benefits in computational efficiency. Other reviewers also find similarity to other continual learning approaches for the graph setting, e.g. ER-GNN. In the author response, the authors point out a number of differences to ER-GNN, such as storing subgraphs instead of nodes for replay, and their coverage maximization strategy focusing on the topology rather than attribute / embedding space.

- Two reviewers questioned whether the method design is appropriate for heterophily graphs, considering some of the method's design aspects are tailored towards homophilic graphs. The author response states some additional strategies for adaptation to heterophily graphs, and that pseudo-training can still benefit this setting (though there is not empirical evidence for this, which is understandable).

- Reviewers also noted that the coverage maximization strategy does not clearly outperform other sampling algorithms, despite the other strategies having much lower coverage ratio, which leads to the question on the importance of the coverage maximization criteria.

In the end, reviewers and AC agree that while the work has promising merits, due to these issues, the work is not yet ready for publication at ICLR. The reviews offer a number of helpful suggestions for improvement, so I encourage the authors to continue improving the paper based on the reviews for future submissions.